# Androgen deprivation induces double-null prostate cancer via aberrant nuclear export and ribosomal biogenesis through HGF and Wnt activation

Won Kyung Kim[1,6], Alyssa J. Buckley[1,6], Dong-Hoon Lee[1], Alex Hiroto[1], Christian H. Nenninger[1], Adam W. Olson [1], Jinhui Wang[2], Zhuo Li[3], Rajeev Vikram[1], Yao Mawulikplimi Adzavon[1], Tak-yu Yau[1], Yigang Bao[1], Michael Kahn[1], Joseph Geradts[4], Guang-Qian Xiao [5] & Zijie Sun [1] ✉

Androgen deprivation therapy (ADT) targeting androgen/androgen receptor (AR)- signaling pathways is the main therapy for advanced prostate cancer (PCa). However, ADT eventually fails in most patients who consequently develop castration-resistant prostate cancer (CRPC). While more potent AR antagonists and blockers for androgen synthesis were developed to improve clinical outcomes, they also show to induce more diverse CRPC phenotypes. Specifically, the AR- and neuroendocrine-null PCa, DNPC, occurs in abiraterone and enzalutamide-treated patients. Here, we uncover that current ADT induces aberrant HGF/MET signaling activation that further elevates Wnt/β-catenin signaling in human DNPC samples. Co-activation of HGF/MET and Wnt/β-catenin axes in mouse prostates induces DNPC-like lesions. Single-cell RNA sequencing analyses identify increased expression and activity of XPO1 and ribosomal proteins in mouse DNPC-like cells. Elevated expression of XPO1 and ribosomal proteins is also identified in clinical DNPC specimens. Inhibition of XPO1 and ribosomal pathways represses DNPC growth in both in vivo and ex vivo conditions, evidencing future therapeutic targets.

Prostate cancer is still the most common malignancy among males and causes about 350,000 deaths worldwide annually[1]. Unlike other human malignancies, the activation of androgen receptor (AR) mediated signaling pathways through binding of androgens is essential for prostate tumorigenesis[2–4]. Almost all primary prostate cancer (PCa) cells express the AR and depend on androgens for their oncogenic growth[3,5]. Therefore, androgen deprivation therapy (ADT) targeting androgen/AR-mediated signaling pathways is the first-line therapeutic option for advanced PCa[6]. However, ADT eventually fails in most patients who consequently develop castration-resistant prostate cancer (CRPC)[2,4]. To inhibit the re-activation of AR-promoted tumor growth via residual androgens[7], more potent AR antagonists and inhibitors for blocking androgen synthesis have been developed in recent years[8]. While these second-generation antagonists/inhibitors showed some effectiveness clinically, emerging evidence has shown that they also induced more diverse CRPC phenotypes[9]. Specifically, a

[1]Department of Cancer Biology and Molecular Medicine, Cancer Center and Beckman Research Institute, City of Hope, Duarte, CA, USA. [2]Integrative Genomics Core, Cancer Center and Beckman Research Institute, City of Hope, Duarte, CA, USA. [3]Electronic Microscopy Core, Cancer Center and Beckman Research Institute, City of Hope, Duarte, CA, USA. [4]Department of Pathology and Laboratory Medicine, Brody School of Medicine, East Carolina University, Greenville, USA. [5]Department of Pathology, Keck School of Medicine, University of Southern California, Los Angeles, CA, USA. [6]These authors contributed equally: Won Kyung Kim, Alyssa J. Buckley. ✉e-mail: zjsun@coh.org

subpopulation of AR- and neuroendocrine (NE)-null PCa cells, double-null PCa (DNPC), has been observed frequently in patients treated with abiraterone (ABI) and enzalutamide (ENZ), directly contributing to the incidence and mortality of metastatic CRPC (mCRPC)[10]. Therefore, understanding the mechanisms for DNPC development is urgently needed for designing specific and effective therapeutic strategies for future treatment.

In this study, we directly address this most urgent and important question in order to identify more effective therapeutic targets for preventing metastatic DNPC development. By analyzing human CRPC datasets, we demonstrate current ADT specifically induces aberrant activation of HGF and MET-mediated signaling pathways, which further elevates Wnt/β-catenin signaling activation, in DNPC cells. In an attempt to mimic the above clinical condition, we develop a series of biologically relevant genetically engineered mouse models (GEMMs) and demonstrate that aberrant activation of HGF/MET and Wnt signaling pathways in the mouse prostate initiates PCa development, promotes disease progression, and induces DNPC-like tumor lesions. Single-cell RNA-sequencing analysis and other in vivo approaches further identify that activated Wnt/β-catenin signaling directly augments SP1-regulated XPO1 expression and increases ribosomal

proteins. Aberrant upregulation of XPO1 and ribosomal protein expression is also identified in human DNPC specimens of patients treated with ABI and ENZ[10]. Using Selinexor, a FDA-approved XPO1 inhibitor[11], we show that XPO1 inhibition significantly represses mouse DNPC-like cell growth in both in vivo and ex vivo conditions, supporting the therapeutic strategies to target XPO1 signaling for treating DNPC.

## Results

### Aberrant activation of HGF/MET and canonical Wnt axes occurs in DNPC cells

To gain in-depth insight for cellular characteristics of DNPC, we analyzed transcriptomics of human CRPC samples[12], defining them as AR pathway active PCa (ARPC), DNPC, and neuroendocrine PCa (NEPC) as previously reported[13]. A significant decrease of AR activated programs by measuring the expression of AR downstream target gene signatures revealed in both DNPC and NEPC compared to ARPC samples (Fig. 1a and Supplementary Fig. 1a). Gene Set Enrichment Analysis (GSEA) using a pre-ranked gene list from differentially expressed genes (DEGs), with a |log$_2$ fold change| > 1 and adjusted *P*-value < 0.05, further identified a significant enrichment in the down-regulation of

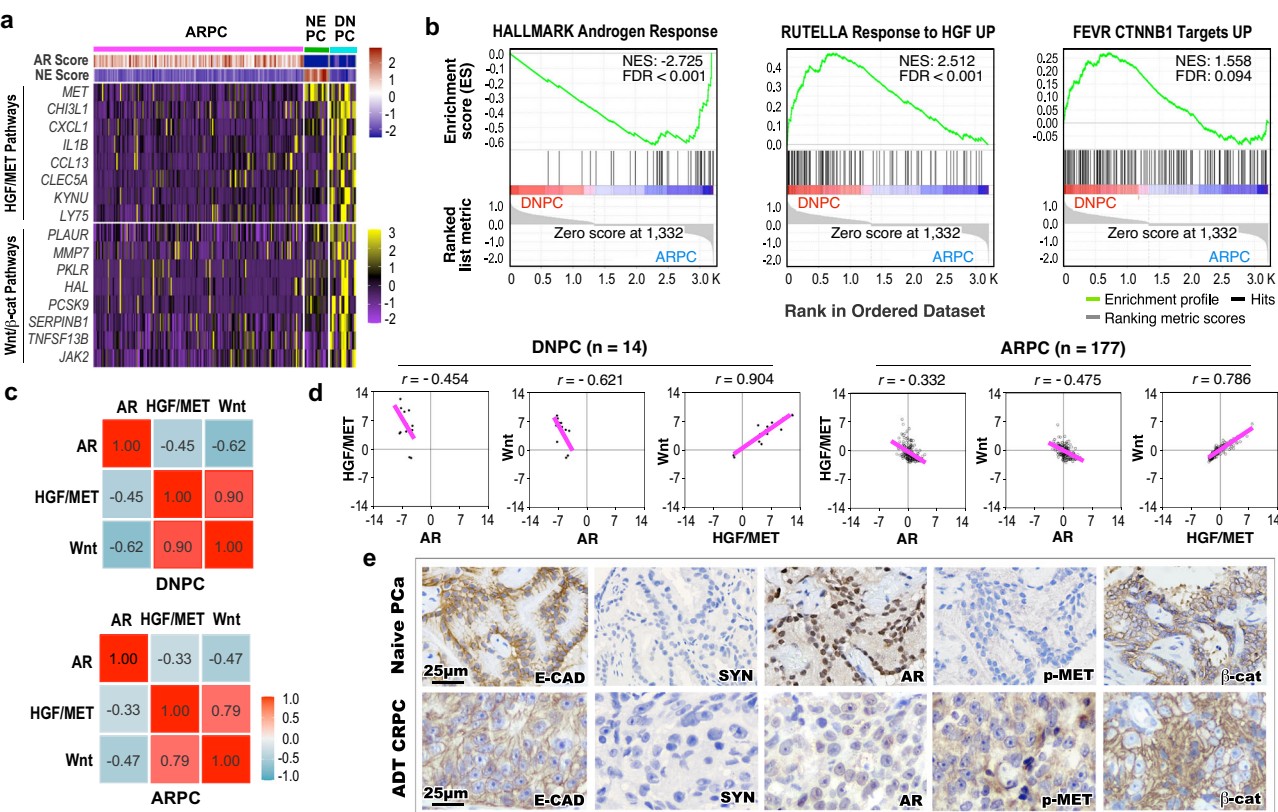

**Fig. 1 | Aberrant activation of HGF/MET and Wnt signaling pathways in human double-negative AR-null NE-null prostate cancer patients. a** Heatmap showing score of indicated gene signatures or expression profile of indicated genes across human metastatic castration-resistant prostate cancer (mCRPC) samples obtained from SU2C/PCF RNA-seq datasets (2019) at cBioPortal[12,13] (ARPC AR-active prostate cancers, *n* = 177; NEPC neuroendocrine prostate cancers, *n* = 13; DNPC double-negative AR-null NE-null prostate cancers, *n* = 14). Scores of AR- or NE-associated gene signature are shown in the top panel based on the previous study[13]. HGF/MET pathway-associated genes and Wnt/β-catenin downstream target genes are listed in the middle and bottom panel, respectively. Colors reflect the level of gene signature score or expression. See "Methods" section. **b** Gene Set Enrichment Analysis (GSEA) enrichment plots of pre-ranked gene list from differentially expressed genes (DEGs) comparing DNPC to ARPC samples. NES normalized enrichment

score, FDR false discovery rate. See Supplementary Data 1. **c** Heatmap of pairwise Spearman correlation between the indicated gene signatures in indicated CRPC samples. Numbers indicate correlation coefficient. Colors reflect the correlation coefficient value. **d** Scatter plots displaying the mRNA expression z-scores of indicated gene signatures in indicated CRPC samples. The pink lines show association between the gene signatures and *r* indicates Spearman's correlation coefficient. **e** Representative images of immunohistochemistry staining using the indicated antibodies on tumor tissues from naive primary prostate adenocarcinoma (PCa, *n* = 5) and androgen deprivation therapy (ADT)-treated CRPC patients (*n* = 6, please also see the "Material" section). AR androgen receptor, NE neuroendocrine, E-CAD E-cadherin, SYN synaptophysin, pMET phosphorylated MET, β-cat β-catenin. Representative images from three independent experiments with similar results are displayed for each micrograph. Scale bars, 25 μm.

androgen-response pathways in DNPC and NEPC in comparison to ARPC samples (Fig. 1b and Supplementary Fig. 1b). However, significant upregulation of HGF/MET and canonical Wnt signaling downstream target genes was observed only in DNPC by comparing transcriptomic changes with ARPC, but not in NEPC with ARPC samples (Fig. 1a). Accordingly, GSEA showed a significant enrichment in the upregulation of HGF/MET and Wnt/β-catenin signaling pathways using the DEGs from DNPC versus ARPC (Fig. 1b and Supplementary Fig. 1c), but not NEPC versus ARPC cells (Supplementary Fig. 1b). Correlation analyses further demonstrated a significant inverse correlation between activation of AR and HGF/MET downstream targets in DNPC samples, directly supporting the above GSEA data (Fig. 1c, d). A significant positive correlation between increased expression of HGF/ MET and Wnt/β-catenin downstream targets was also observed in DNPC as well as ARPC samples (Fig. 1c, d). Representative images of immunohistochemical analyses (IHC) showed E-cadherin positive and synaptophysin (SYN) negative staining in PCa cells of both naive primary and ABI- and ENZ-treated CRPC patient samples. However, lack of typical nuclear AR staining along with positive phosphorylated MET (pMET) and cytoplasmic β-catenin staining was observed in adjacent sections of the above CRPC tissues but not in primary PCa samples (Fig. 1e, please see the "Methods" section), reaffirming activating MET and β-catenin signaling pathways correlated with reduced nuclear AR expression in those human CRPC cells. To gain in-depth insight into dysregulation of MET and Wnt/β-catenin axes during CRPC development, we analyzed single-cell RNA-sequencing (scRNA-seq) datasets derived from human primary PCa and mCRPC[14]. After integrating the samples, epithelial cells were extracted and re-clustered, and comparable cell clusters were identified and verified with NE and AR scores in both primary PCa and mCRPC samples (Supplementary Fig. 1d–h). The luminal epithelial cell clusters 6–8 (LE6-8), predominately existed in mCRPC samples with significantly lower AR scores than other clusters mainly identified in primary PCa samples (Supplementary Fig. 1h). Violin expression plots also showed low expression of AR in these clusters (Supplementary Fig. 1i). In contrast, increased expression of *MET*, *CTNNB1*, and their downstream targets, including *SERPINB1*, *CLDN1*, *DKK1*, *CCND1*, *MMP7*, and *PLAUR*, respectively, was detected in two mCRPC predominant clusters, LE7 and LE8 (Supplementary Fig. 1i). These scRNA-seq data provide high-resolution depictions to directly link reduced AR expression and activity with increased activation of *MET* and *CTNNB1* signaling pathways in mCRPC cells. Taken together, our data identified a correlation between increased HGF/ MET and Wnt/β-catenin activation with reduced AR signaling pathways, which uncovers a distinct mechanism for DNPC development.

### Reciprocal activation of HGF/MET signaling activates canonical Wnt signaling in prostate oncogenesis

To gain a direct and mechanistic insight into the activation of HGF/ MET to induce Wnt/β-catenin signaling in DNPC pathogenesis, we generated *hHGFtg:H11^hMET/+^:PB^Cre4^* double transgenic mice, referred to as DoubleTg (Fig. 2a). These mice have concomitant ubiquitous expression of human *HGF* transgene (*hHGFtg*) and conditional expression of human *MET* transgene (*hMETtg*) to mimic the paracrine interaction between aberrant HGF and MET activation as observed in PCa patients[15–17]. Elevated HGF levels were observed specifically in sera of DoubleTg compared to *H11^hMET/+^:PB^Cre4^* mice (Fig. 2b). Accordingly, robust pMET was detected in atypical cells within prostatic intraepithelial neoplasia (PIN) lesions of DoubleTg mice but not in *H11^hMET/+^:PB^Cre4^* mice despite the expression of transgenic human MET appearing in atypical cells of both samples (Fig. 2c). Additionally, no expression of MET and pMET was detected in prostate tissues of wild-type controls (Supplementary Fig. 2a). Based on the guidelines provided by "the New York Pathology Panel"[18], we identified pathological lesions representing low and high-grade PIN (LGPIN, HGPIN), or intracystic adenocarcinomas in prostate tissues of 3-, 6-,

or 9-month-old DoubleTg mice, respectively, (Fig. 2d and Supplementary Fig. 2b), demonstrating reciprocal activation of MET via HGF promoting PCa development. To assess the regulatory mechanism underlying the reciprocal activation of HGF/MET in prostate tumorigenesis, we performed scRNA-seq analyses using pathologically confirmed prostate tissues containing HGPIN and PCa lesions from DoubleTg mice. After quality controls and filtering procedures, cell sets were viewed in Uniform Manifold Approximation and Projection (UMAP) plots[19–21] (Fig. 2e and Supplementary Fig. 2c–g). To assess the in-depth molecular changes in prostatic tumor epithelia, we extracted epithelial cells and re-clustered them (Supplementary Fig. 2h–j). The expression of *hMETtg* appeared mainly in LE clusters (Fig. 2f and Supplementary Fig. 2k). The DEGs, which showed in more than 5% of cells with an adjusted *P*-value < 0.05 and |average log$_2$ fold change| > 0.1, were identified using a Wilcoxon Rank Sum test by comparing *hMETtg*+ and *hMETtg*− epithelial cells (Fig. 2g). GSEA using the above pre-ranked DEGs identified a significant enrichment in upregulation and activation of Wnt/β-catenin and related Myc activation pathways (Fig. 2h), as well as pathways directly related to prostate tumorigenesis (Supplementary Fig. 2l). A significantly higher expression of *hMETtg* and β-catenin downstream targets, *Cd44*, *Sox9*, *Tcf7l2*, *Mmp7*, and *Plaur* was also identified in *hMETtg*+ than *hMETtg*− epithelial cells on violin expression plots (Fig. 2i). Spearman gene-gene correlation analysis further demonstrated a significant correlation of the *hMETtg* with those β-catenin targets (Fig. 2j). Quantitative reverse transcription-PCR (qRT-PCR) analyses verified the higher expression of Wnt/β-catenin targets, *Cd44*, *Sox9*, *Mmp7*, and *Plaur* in RNA samples prepared from PCa tissues of DoubleTg mice than those from controls (Fig. 2k). IHC analyses revealed an increase in cytoplasmic and nuclear β-catenin expression in PCa cells with positive pMET expression in adjacent tissue sections (Red arrows in the top panel, Fig. 2l) but not in prostate samples of wild-type control mice (the left panel, Supplementary Fig. 2m). Accordingly, co-immunofluorescence (co-IF) analyses also showed overlay of cytoplasmic and nuclear β-catenin with pMET staining in these prostate tissue specimens (blue arrows in the bottom panel, Fig. 2l), and there is a significant increase in double positive cells compared to prostate tissues from the wild-type mice (the right panel, Supplementary Fig. 2m, n). These lines of experimental evidence implicate a regulatory mechanism by which activating HGF/MET signaling induces canonical Wnt axes during the course of PCa development.

### Aberrant HGF/MET and canonical Wnt activation develops robust invasive and metastatic PCa with double-negative cellular properties

To directly examine the biological consequences of HGF/MET and canonical Wnt signaling co-activation in prostate epithelia, we developed the *hHGFtg:H11^hMET/+^:Ctnnb1^L(Ex3)/+^:PB^Cre4^* mouse model, referred to as TripleTg, in which both the human *MET* transgene and stabilized β-catenin are expressed in prostatic epithelia in combination with ubiquitous human *HGF* transgene expression to mimic observations from clinical DNPC samples (Fig. 3a). Elevated levels of HGF were also detected in sera of TripleTg mice (Supplementary Fig. 3a). Robust invasive prostate adenocarcinomas developed much earlier in TripleTg than DoubleTg, or stabilized β-catenin only transgenic mice (Fig. 3b). Moreover, metastatic tumor lesions at multiple organs revealed as early as at 9 months of age in TripleTg mice (Fig. 3b), corresponding to the shortest survival rate compare with other control mice (Fig. 3c). The faster growing and more aggressive tumor behaviors were identified grossly in representative 6-, and 10-month-old TripleTg mice, featuring multiple tumors in different prostatic lobes, local invasion into seminal vesicles, and multiple metastatic tumor lesions (Fig. 3d). Pathological examination of 6-month-old TripleTg mice showed invasive adenocarcinoma lesions with positive staining for both pMET and nuclear β-catenin in the malignant prostate tissue

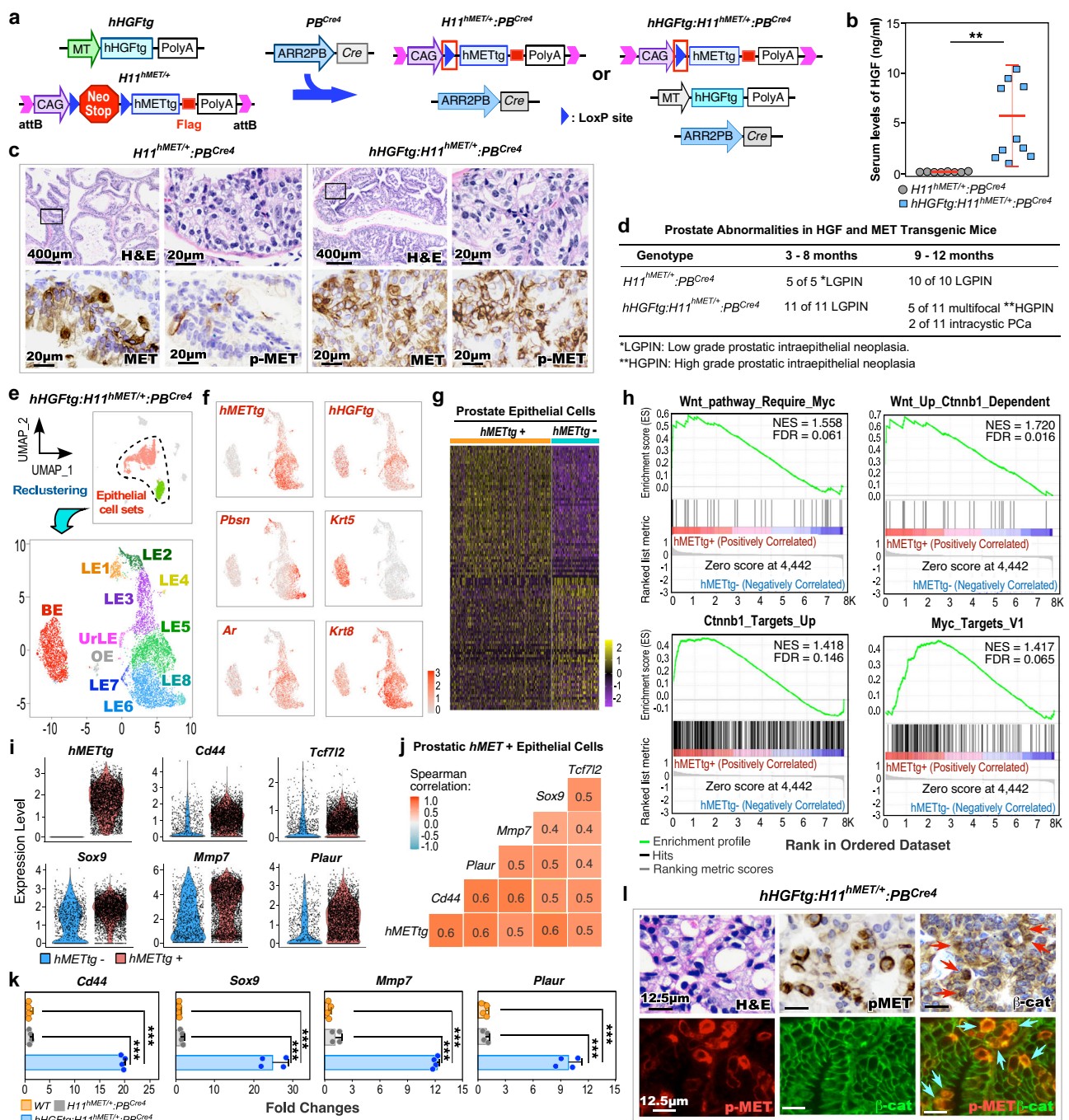

**Fig. 2 | HGF/MET signaling activates canonical Wnt signaling in prostate oncogenesis. a** Schematic of the human HGF (*hHGFtg*) and MET (*H11^hMET*) transgenes, and *PB^Cre4* alleles, shown in relation to the mating strategy. **b** Serum HGF levels of *H11^hMET/+:PB^Cre4* (*n* = 8) and *hHGFtg:H11^hMET/+:PB^Cre4* (*n* = 10) mice. **c** Representative images of hematoxylin-eosin (H&E) and immunohistochemistry (IHC) staining using the indicated antibodies on adjacent prostate tissues from the indicated mice. Scale bars, 400 µm, 20 µm. **d** Table summarizing the pathological abnormalities in the prostates of *H11^hMET/+:PB^Cre4* and *hHGFtg:H11^hMET/+:PB^Cre4* mice at the indicated age. **e** Uniform Manifold Approximation and Projection (UMAP) plots presenting total cells (*n* = 9236) with highlighting prostatic epithelial cells (both green and red cell clusters) from *hHGFtg:H11^hMET/+:PB^Cre4* mice, and epithelial cells (*n* = 7286) being further sub-clustered, re-clustered, and labeled by epithelial cell cluster (bottom). The dotted line delineated the prostatic epithelial cells. BE basal epithelial cells, LE luminal epithelial cells, UrLE urethral epithelial cells, OE other epithelial cells. **f** Gene expression UMAP plots for the indicated genes in epithelial cells (*n* = 7286). Color intensity indicates the scaled expression level. **g** Heatmap

showing DEGs between *hMETtg+* and *hMETtg−* epithelial cells. See Supplementary Data 2. **h** GSEA plots showing the positive enrichment of the indicated gene sets comparing *hMETtg+* and *hMETtg−* cells. NES normalized enrichment score, FDR false discovery rate. **i** Violin plots visualizing the expression levels of *hMETtg* and Wnt downstream target genes in *hMETtg+* (*n* = 4533) and *hMETtg−* (*n* = 2753) epithelial cells. Black dots correspond to individual epithelial cells. **j** Heatmap of pairwise Spearman correlation between the indicated genes in *hMETtg+* epithelial cells. Colors reflect the correlation coefficient value. **k** qPCR analysis of the indicated genes shown as fold change in indicated mouse prostate tissues from four biological replicates. **l** Representative images of H&E, IHC, and immunofluorescence staining (IF) using indicated antibodies on adjacent sections from *hHGFtg:H11^hMET/+:PB^Cre4* mice. Red and blue arrows indicate nuclear β-catenin and co-overlay of pMET with nuclear β-catenin, respectively. Representative images with consistent results from three biological replicates are shown. Scale bars, 12.5 µm. In **b** and **k**, data are mean ± s.d. **P < 0.01, ***P < 0.001; Unpaired two-tailed *t*-tests. See source data and the exact P values in the Source Data file.

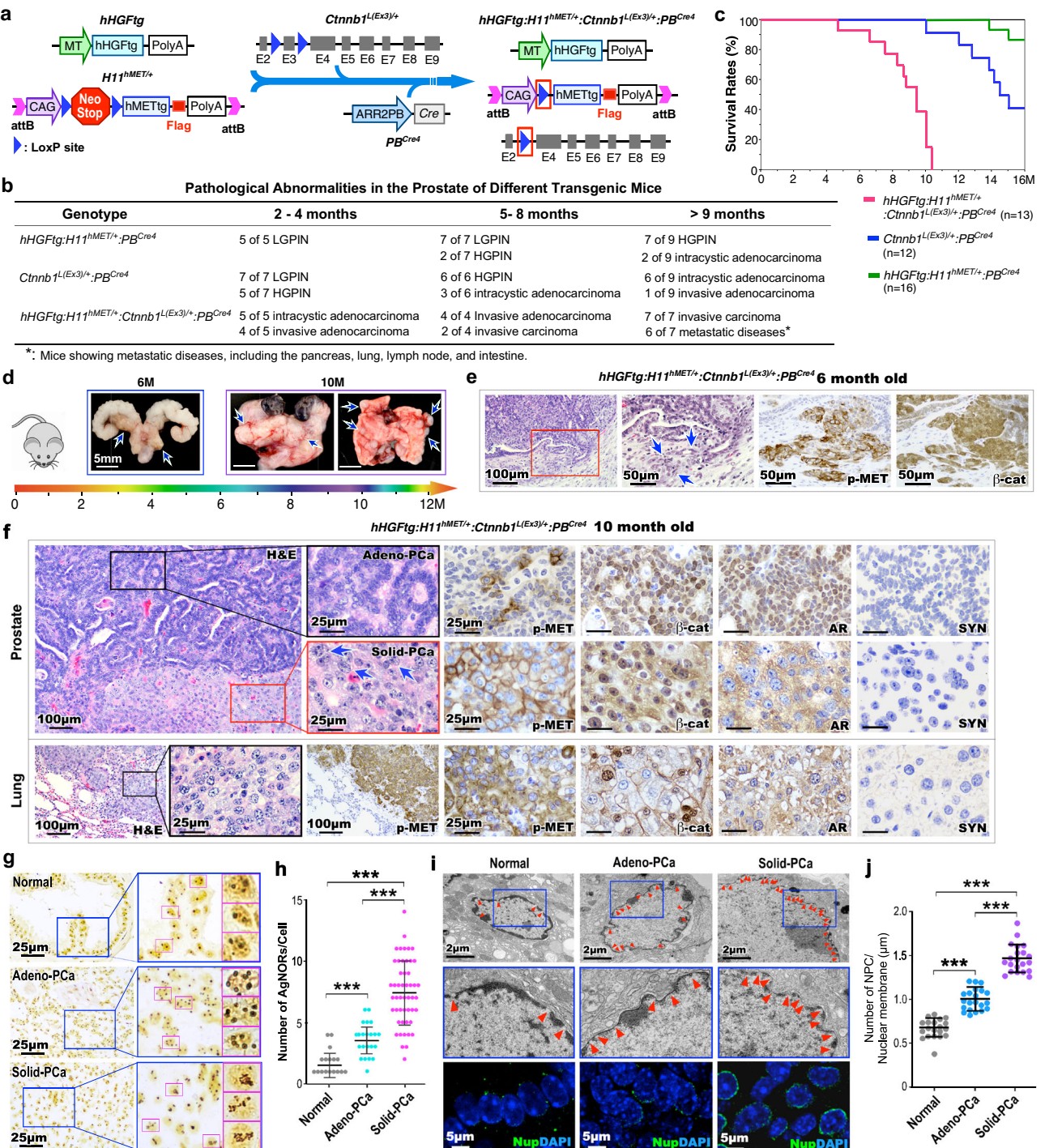

**Fig. 3 | Aberrant activation of HGF/MET and Wnt signaling pathways develops DNPC with metastatic and aggressive properties. a** Schematic of generating different transgenic mice as indicated above. **b** Table summarizing the pathological abnormalities in the prostates from different genotype mice. **c** Kaplan–Meier survival curves for the indicated transgenic mice. **d** Representative gross images of prostate tumor tissues with seminal vesicles and urinary bladders and lung tissues from either 6- or 10-month-old *hHGFtg:H11^{hMET/+}:Ctnnb1^{L(Ex3)/+}:PB^{Cre4}* mice. Blue arrows indicate primary tumor or metastasis loci. **e** Representative images of H&E and IHC staining using the indicated antibodies on adjacent prostate tissues from *hHGFtg:H11^{hMET/+}:Ctnnb1^{L(Ex3)/+}:PB^{Cre4}* mice at 6 months of age. Blue arrows point to invasive lesions. **f** Representative images of H&E and IHC staining using the indicated antibodies on adjacent prostate (top) and lung (bottom) tissues from 10- month-old *hHGFtg:H11^{hMET/+}:Ctnnb1^{L(Ex3)/+}:PB^{Cre4}* mice. Blue arrows indicate enlarged nucleoli. **g** Representative images of AgNOR stained prostate tissue sections containing normal prostatic glands, gland-forming prostate adenocarcinoma (Adeno-PCa), and

solid prostate carcinoma (Solid-PCa) lesions in *hHGFtg:H11^{hMET/+}:Ctnnb1^{L(Ex3)/+}:PB^{Cre4}* mice. Pink box highlights single nucleus. **h** Quantification of AgNOR number in individual cells from the indicated loci. Numbers of AgNORs per cells were measured for 50 cells in each image, and at least 5 images from three biological replicates were analyzed. **i** Representative images of transmission electron microscopy (top) and immunofluorescent staining for Nups (NUP62, NUP98, and NUP214) detected by mAb414 antibody (bottom) in the indicated loci from prostate tissues. Red arrowheads point to nuclear pore complexes (NPC). Nuclei were stained with DAPI. **j** Numbers of NPC per μm of nuclear membrane were quantified, at least 20 cells in each image and 5 images from three biological replicates were analyzed. Representative images with consistent results from three biologically independent experiments are shown. Scale bars, 100 μm (**e**, **f**), 50 μm (**e**), 25 μm (**f**, **g**), 5 μm (**i**), 2 μm (**i**). In **h** and **j**, data are represented as mean ± s.d. **P < 0.01; Unpaired two-tailed *t*-tests. Source data and the exact *P* values are provided in the Source Data file.

(Fig. 3e). Analyses of prostate tissues from 10-month-old TripleTg mice identified a series of progressive tumor lesions coexisted in the same prostate tissues. They included lesions with glandular characteristics and well-differentiated adenocarcinomas, similar to human Gleason Grade 3–4 prostate carcinomas, termed "Adeno-PCa" in this study, as well as lesions with poorly differentiated characteristics containing abundant lightly eosinophilic cytoplasm and pleomorphic nuclei without distinct gland formation, akin to Gleason Grade 5 prostate carcinomas, termed "Solid-PCa" (Fig. 3f). It should be noted that both Adeno- and Solid-PCa are terms only used to denote the phenotypes in this study. Solid-PCa cells additionally displayed abnormal nucleolar characteristics with prominent and often multiple nucleoli (blue arrows, Fig. 3f). Positive pMET and nuclear β-catenin staining were observed in both types of tumor cells. However, typical nuclear AR staining only appeared in Adeno-PCa cells. In contrast, Solid-PCa cells showed noticeable cytoplasmic but no or very weak nuclear AR staining (Fig. 3f), presenting with similar cellular properties to those observed in human DNPC (Fig. 1e). Both types of tumor cells also revealed positive staining for CK8 and E-cadherin but were negative for SYN verifying their epithelial properties (Fig. 3f and Supplementary Fig. 3b). Analysis of lung metastatic tumor lesions showed similar cell characteristics as observed in poorly differentiated Solid-PCa, featuring lightly eosinophilic cytoplasm and irregular nuclei with distinctly permanent nucleoli (Fig. 3f). Positive staining for pMET and nuclear β-catenin, with no nuclear AR staining was also detected in lung metastatic tumor cells (Fig. 3f). Gross and histological examination of metastatic tumor lesions from intestine and spleen tissues revealed very similar tumor cell characteristics as found in the lung metastatic lesions (Supplementary Fig. 3c). Measuring the expression of AR and SYN in human PCa (Fig. 1e) and TripleTg mouse prostate PCa tissues (the middle panel, Fig. 3f) showed significant reduction of nuclear AR expression in both PCa cells in human tissues isolated from patients treated with ABI and ENZ and Solid-PCa cells from TripleTg mice (Supplementary Fig. 3d). Taken together, development of aggressive and metastatic PCa lesions in TripleTg mice demonstrates a significant role for co-activation of HGF/MET and canonical Wnt axes in promoting PCa progression. Specifically, the lack of nuclear AR expression in poorly differentiated Solid-PCa cells of TripleTg mice corroborates the cellular characteristics observed in human DNPC cells, further suggesting a regulatory role of aberrant activation of HGF/MET and Wnt axes in PCa progression and DNPC development.

Increasing nucleolar size and number, resulting from elevated ribosome synthesis, are considered as hallmarks of aggressive tumor cells and are closely correlated to poor prognosis[22]. To assess the nucleolar abnormalities identified in PCa cells of TripleTg mice (Fig. 3f), we performed AgNOR assay to examine silver-stained proteins associated with the nucleolar organizer regions, AgNORs, in the nuclei. Significantly more AgNORs revealed in Solid-PCa than Adeno-PCa, or normal prostatic epithelial cells, correlating to a larger nucleolar area (Fig. 3g, h and Supplementary Fig. 3g). Because the size and number of AgNORs reflect nucleolar and cell proliferative activity of tumor cells[23], these results further correlated with the fast-growing and aggressive characteristics of Solid-PCa cells. Using the transmission electronic microscopy (TEM) analyses, we identified much larger nuclear size in Solid-PCa than in Adeno-PCa cells and normal epithelial cells (Supplementary Fig. 3e, f). Examining clinical samples also showed much higher number of AgNORs and larger size of nucleoli in CRPC cells of ABI- and ENZ-treated patient samples than naïve PCa cells (Supplementary Fig. 3h, i), demonstrating similar cellular properties between mouse Solid-PCa and human DNPC cells.

Because the nuclear pore complexes (NPC) function as the central mediators of nucleo-cytoplasmic transport, increased numbers of NPC amplify the nuclear transport machinery to promote tumor growth, and frequently occur in aggressive tumors[24,25], including PCa cells[26]. TEM analyses showed significantly more NPC in Solid-PCa cells than

Adeno-PCa and control epithelial cells (Fig. 3i, j). IF assays revealed increased expression of nucleoporins (Nups), the structural components of the NPC, in Solid-PCa cells in comparison to other control samples (Fig. 3i). Identifying these multiple nuclear abnormalities in Solid-PCa cells directly supports their aggressive and metastatic tumor cellular characteristics and provides in-depth molecular and cellular changes induced by co-activating HGF/MET and Wnt signaling to promote PCa progression and DNPC development.

## Aberrant HGF/MET and canonical Wnt activation induces nuclear exporting and ribosomal synthesis pathways to promote PCa progression and metastasis

To understand the regulatory role of HGF/MET and canonical Wnt axes in PCa progression and metastasis, we performed scRNA-seq analyses using pathologically confirmed PCa tissues of TripleTg mice. The scRNA-seq samples from TripleTg and DoubleTg mice were integrated, aligning 9 similar cell subsets from both samples based on their transcriptomic profiles[19] (Supplementary Fig. 4a–d). To gain high resolution of the cellular properties, epithelial cells were extracted from total cells, and re-clustered to 15 epithelial cell clusters following cell cycle regression (Fig. 4a and Supplementary Fig. 4e–h). UMAP expression plots showed similar Ar expression in epithelial cells of TripleTg and DoubleTg mice, but with noticeable reduction of AR-regulated gene, e.g. Pbsn and Fkbp5, expression, and elevated expression of β-catenin downstream targets, Axin2 and Tcf4, revealed in TripleTg samples in comparison to DoubleTg samples (Fig. 4b). GSEA using DEGs of hMETtg+ cells of TripleTg versus those of DoubleTg samples further showed down-regulation of androgen signaling pathways with upregulation of Wnt/β-catenin signaling pathways (Fig. 4c and Supplementary Fig. 4i), consistent with our earlier observation of reduced nuclear AR and increased stabilized β-catenin expression in Solid-PCa cells of TripleTg mice (Fig. 3f). Additionally, pathways related to ribosome synthesis, nuclear exporting, and tumor metastasis were also significantly enriched in TripleTg samples (Fig. 4c and Supplementary Fig. 4i). Violin expression plots showed reduced Ar expression, decreased expression of Pbsn and Fkbp5, and higher expression of Axin2 and Tcf4, and Xpo1, a nuclear exporting regulator, Rpl12 and Rpl16, ribosomal proteins, in hMETtg+ cells of TripleTg mice than those of DoubleTg counterparts (Fig. 4d). Using qRT-PCR analyses further showed increased expression of Xpo1, Rpl12, Rps16, and Eif4a1 in RNA samples from primary prostate and lung metastatic tumor cells of TripleTg mice in comparison to those from PCa tissues of DoubleTg mice (Fig. 4e). These data taken together demonstrate the regulatory role of HGF/MET and canonical Wnt co-activation in increasing nuclear exporting and ribosomal synthesis pathways in hMETtg+ tumor cells of TripleTg mice, which also correlate to the nuclear abnormalities associated with ribosomal synthesis observed in Solid-PCa cells[22,23].

Analyzing integrated epithelial cell clusters showed that BE4 and LE1–4 clusters were predominant in TripleTg samples, but other clusters were enriched in DoubleTg samples (Supplementary Fig. 4j). Reduced expression of AR downstream targets, but increased expression of HGF/MET and Wnt/β-catenin downstream target genes as well as ribosomal related genes were mainly identified in BE4 and LE1–4 clusters by transcriptomic analyses across the epithelial cell clusters (Fig. 4f and Supplementary Fig. 4k). Additionally, increased expression of genes directly related to cell proliferation were observed specifically in LE3 and LE4 clusters, providing the molecular basis for the poorly differentiated and fast-growing tumor cells characteristics observed in prostate tumors of TripleTg mice. Using single-cell trajectory analyses by Monocle3, we assessed dynamic and in-depth transcriptomic changes governing tumor development and progression in hMETtg+ cells of TripleTg mice[27]. As shown in pseudotime trajectory plots (Fig. 4g), BE4 cells act as a starting point and further differentiate and progress to luminal cell branches mainly constituting LE2, 3, and 4 clusters. Significantly low expression of Pbsn and Fkbp5,

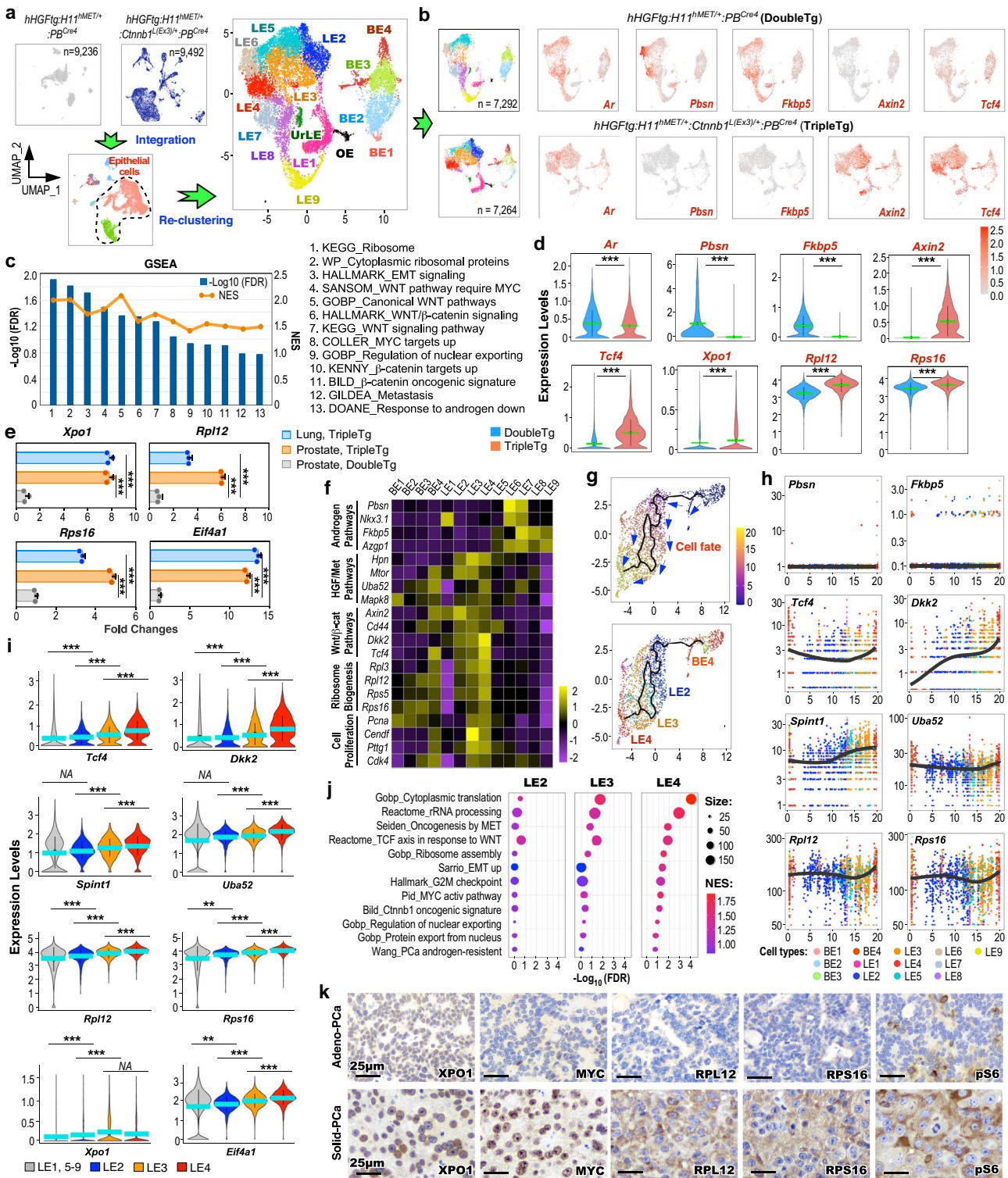

with gradually elevated expression of *Tcf4* and *Dkk2*, *Spint1* and *Uba52*, HGF/MET signaling downstream targets, and *Rpl12* and *Rps16*, respectively, were revealed through BE4 to LE2-4 cell clusters during pseudotime progression (Fig. 4h). Violin expression plots further showed gradually increased expression of *Tcf4*, *Dkk2*, *Spint1*, *Uba52*, *Rpl12*, *Rps16*, *Xpo1*, and *Eif4a1* through LE2, 3, and 4 clusters in comparison to other LE clusters of TripleTg samples (Fig. 4i). GSEA using DEGs of LE2, 3, or 4 versus other LE clusters identified a significant enrichment in the signaling pathways related to protein synthesis, translation, rRNA processing, HGF/MET and Wnt signaling activation, nuclear export, ribosome biogenesis, and epithelial-mesenchymal

transition (EMT) activation (Fig. 4j). These data provide a dynamic, single-cell resolution depiction of aberrant co-activation of HGF/MET and Wnt/β-catenin in elevating nuclear exporting, ribosome synthesis, and oncogenic pathways to promote PCa progression and DNPC development. In contrast, *hMETtg+* cells from DoubleTg mice exhibited a different trajectory fate, starting with BE1–3 cells and differentiating into luminal cell branches possessing LE5–9 cells (Supplementary Fig. 4l). Additionally, increased expression of AR-regulated genes and reduced expression of HGF/MET and Wnt/β-catenin downstream targets as well as *Rpl12* and *Rps16* revealed through luminal *hMETtg+* cells of DoubleTg mice (Supplementary

**Fig. 4 | Nuclear export signal and ribosomal biogenesis promote prostate cancer progression and aggressiveness. a** Individual UMAP visualization of cells from DoubleTg (top left, gray) and TripleTg (top middle, dark blue) prostates, and integrated cells colored by cell type identities (bottom). UMAP plot of epithelial cells re-clustered and labeled as BE, basal epithelial cells; LE, luminal epithelial cells; UrLE, urethral epithelial cells; OE, other epithelial cells (top right). **b** UMAP plots showing indicated gene expression, separated by each genotype. Color intensity indicates the scaled expression level. **c** GSEA compares *hMETtg*+ cells from TripleTg versus DoubleTg mice. See Supplementary Data 3. **d** Violin plots visualizing the indicated gene expressions in *hMETtg*+ cells (DoubleTg, *n* = 4417; TripleTg, *n* = 2144) of each genotype. **e** qPCR analysis of the indicated genes shown as fold changes in the indicated tissues from three biological replicates. **f** Heatmap depicting average expression of genes associated with indicated pathways in each cluster. **g** Pseudotime trajectory plots of *hMETtg*+ cells (*n* = 2144) from TripleTg

mice, visualized on UMAP plots by pseudotime (top) and cluster identity (bottom). **h** Linear pseudotime expression plots showing dynamics of indicated gene expression over pseudotime in *hMETtg*+ cells from TripleTg mice. Dots correspond to individual cells colored by cluster identity. **i** Violin plot visualizing the expression levels of indicated genes from TripleTg mice (LE1, 5–9, *n* = 1934; LE2, *n* = 1165; LE3, *n* = 1666; LE4, *n* = 1096). **j** Bubble charts showing GSEA by comparing LE2, LE3, or LE4 versus other LEs in TripleTg sample. Color and size of bubbles represents NES and weighted numbers of genes. See Supplementary Data 4–6. **k** Representative images of IHC using indicated antibodies on adjacent sections from TripleTg mice. Representative images with consistent results from three biological replicates are shown. Scale bars, 25 μm. In **d–i**, data are mean ± s.d. \*\**P* < 0.01 and \*\*\**P* < 0.001, two-tailed Wilcoxon Rank Sum tests (**d, i**), unpaired two-tailed *t*-tests (**e**). Green or blue bar indicates the mean value (**d, i**). Source data and the exact *P* values are provided in the Source Data file.

Fig. 4m). IHC analyses using adjacent PCa tissues of TripleTg mice showed the higher expression of XPO1, MYC, RPL12, RPS16, and pS6 in Solid-PCa cells than Adeno-PCa cells (Fig. 4k), further supporting the above scRNA-seq results, and suggesting that Solid-PCa are derived from Adeno-PCa during tumor progression in TripleTg mice. Given that MYC is a master regulator of ribosome biogenesis[28], identifying increased MYC expression in Solid-PCa cells through activating Wnt/β-catenin signaling outlines a regulatory mechanism for aberrant activation of XPO1, ribosomal biogenesis, and protein synthesis pathways to promote PCa progression and DNPC development. Analyzing clinical samples, we also identified elevated expression of XPO1 and RPL12 in DNPC samples from ABI- and ENZ-treated patients (Supplementary Fig. 5a). An increase in the expression of *XPO1* was also detected in two human mCRPC predominant clusters, LE7 and LE8 (Supplementary Fig. 5b)[14], which has been shown to possess activated HGF/MET and Wnt/β-catenin axes (Supplementary Fig. 1i). Analyses of recent human CRPC datasets identified significantly reduced AR downstream targets (AR score) in DNPC in comparison to ARPC samples (Supplementary Fig. 5c)[29]. Accordingly, higher expression for HGF/MET downstream targets (HGF score) and WNT/β-catenin downstream targets (WNT score) was also revealed in DNPC samples. Significantly reduced AR expression and increased XPO1 expression were further shown in DNPC in comparison to ARPC (Supplementary Fig. 5d). Taken in total, these lines of experimental evidence further support that aberrant co-activation of HGF and Wnt/β-catenin signaling pathways increase XPO1 expression during DNPC development.

## Aberrant activation of HGF and Wnt signaling increases CRM1/XPO1 expression through increased SP1 expression

Identifying elevated expression of XPO1 in poorly differentiated Solid-PCa cells in TripleTg mice suggests a regulatory role of Wnt/β-catenin signaling in PCa progression and DNPC development. It has been shown that SP1 regulates *Xpo1* transcription[30] and β-catenin enhances SP1 transcriptional activity through directly interacting and stabilizing the SP1 protein[31]. Using IHC approaches, we first assessed the expression of stabilized β-catenin on SP1 and XPO1 expression in mouse PCa samples. While the expression of cytoplasmic and nuclear β-catenin appeared in Adeno-PCa cells, prominent nuclear β-catenin expression was observed in both Solid-PCa and lung metastatic tumor cells of TripleTg mice (Fig. 5a). Accordingly, increased expression of SP1 and peri-nuclear staining of XPO1 only showed in both prostatic Solid-PCa and lung metastatic tumor cells but not in Adeno-PCa cells of TripleTg mice and PCa samples of DoubleTg mice in adjacent tissue sections (Fig. 5a and Supplementary Fig. 5e). Co-IF analyses further showed that elevated expression of SP1 was overlaid with extensive stabilized β-catenin expression in Solid-PCa cells but not in Adeno-PCa cells of TripleTg mice, and PCa cells of DoubleTg mice (Fig. 5a, b and Supplementary Fig. 5f). Moreover, predominant peri-nuclear staining of XPO1, overlaying with the nuclear SP1, was specifically seen in Solid-PCa cells but not in Adeno-PCa cells and in PCa cells of DoubleTg mice

(Fig. 5a, b and Supplementary Fig. 5f). These data elucidate the regulatory role of stabilized β-catenin on increasing SP1 and XPO1 expression in Solid-PCa cells. Multiple SP1 binding sites were identified within the promoter region of the *Xpo1* gene and, through these sites, SP1 can regulate *Xpo1/Crm1* transcription in transformed tumor cells[30]. Using chromatin immunoprecipitation-quantitative PCR (ChIP-qPCR) analyses, we directly examined the binding of SP1 to the promoter of *Xpo1* thereby activating its transcription. Specific occupancy of SP1 was identified in the SP1 binding sites within both the promoter and enhancer regions of the *Xpo1* gene locus, but not the *Untr4* locus, used as a negative control, in PCa cells isolated from TripleTg mice, and also not in PCa cells isolated from DoubleTg mice (Fig. 5c)[30]. Using knockdown approaches, we further demonstrated that specific reduction of SP1 expression significantly diminishes the expression of XPO1 transcripts and proteins in two human prostate cancer cell lines, DU145 and PC3, as well as in prostate organoid cultures derived from PCa tumors of TripleTg mice (Fig. 5d, e). These lines of experimental evidence demonstrate the regulatory role of Wnt/β-catenin signaling activation in enhancing SP1-mediated *Xpo1* expression in PCa cells. Given that dysregulation of XPO1 directly contributes to tumor development, progression, and cancer drug resistance[32–34], our findings implicate a regulatory mechanism underlying aberrant Wnt/β-catenin activation to promote PCa progression, hormone refractoriness, and DNPC development through enhancing SP1-regulated XPO1 activation.

## Aberrant activation of XPO1 and ribosomal synthesis pathways converts PCa cellular properties and promotes androgen-independent growth

Our data from scRNA-seq and other approaches identified aberrant XPO1 and ribosomal synthesis activation in Solid-PCa cells from TripleTg mice. Using both organoid culture and in vivo graft approaches, we directly assessed these abnormalities in regulating androgen-independent PCa cell growth (Fig. 6a). Intriguingly, prostatic tumor organoids that were derived and developed from dissected PCa cells of TripleTg mice showed the ability to grow in culture either with or without dihydrotestosterone (DHT) (Fig. 6b). Histological analyses recapitulated similar cellular characteristics of Adeno-PCa in organoids cultured with DHT, however, their counterparts cultured without DHT showed less differentiated tumor characteristics, similar to Solid-PCa cells (Fig. 6b). Organoids cultured with DHT showed positive nuclear AR staining but those cultured without DHT showed no nuclear AR staining with correspondingly more intense peri-nuclear staining of XPO1 and stronger cytoplasmic staining of RPS16, whereas both samples revealed positive E-cadherin staining (Supplementary Fig. 5g). The AR antagonist, ENZ demonstrated significant inhibition of tumor organoid growth only in samples cultured with DHT but not in those cultured without DHT (Fig. 6c). However, Selinexor or CX5461, a XPO1 or ribosome inhibitor, respectively, displayed a significant repressive effect on organoid growth in both samples cultured with or

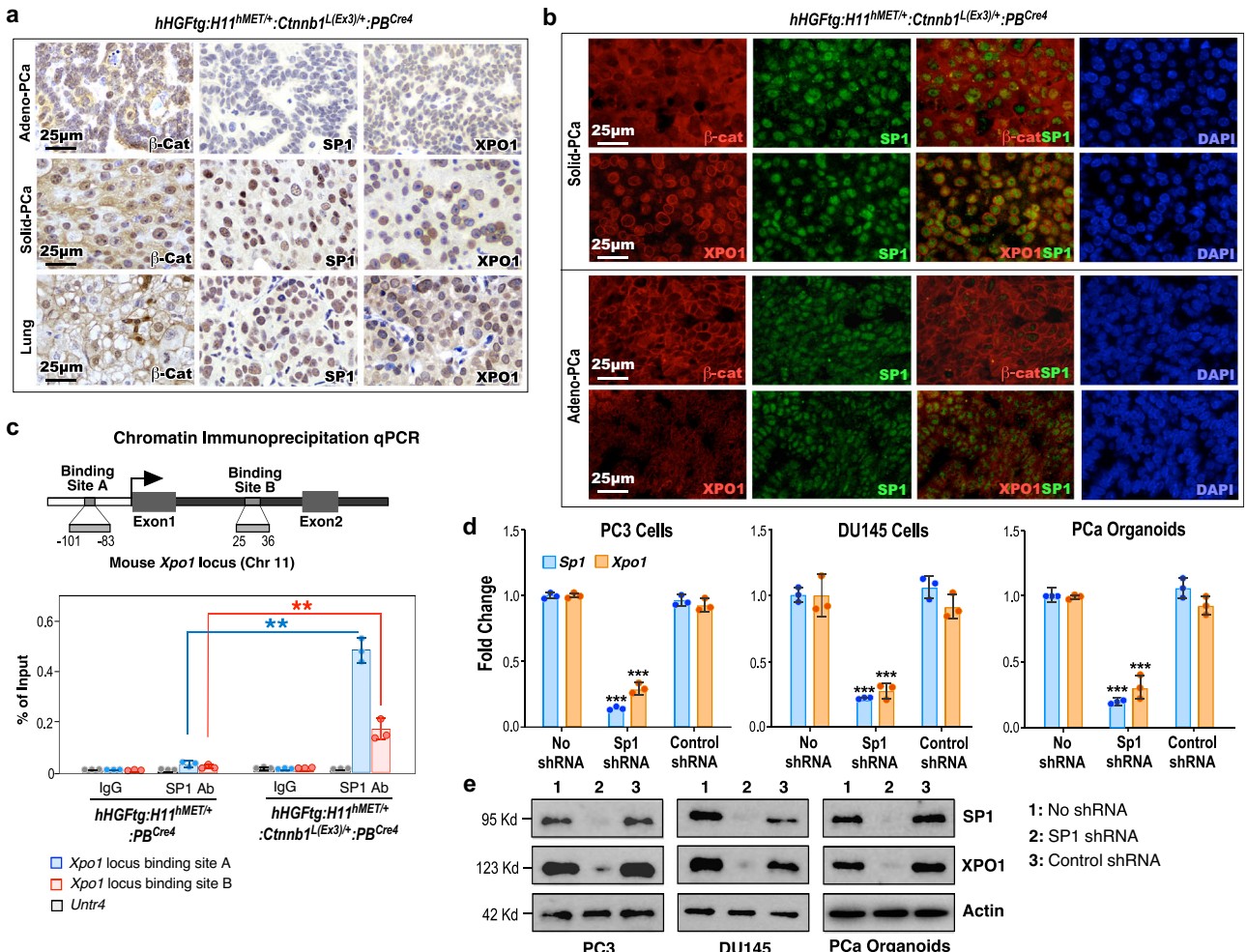

**Fig. 5 | Aberrant activation of HGF and Wnt signaling increases CRM1/XPO1 expression through increased SP1 expression. a** Representative images of IHC staining using the indicated antibodies on different prostate lesions (top and middle) and lung (bottom) tissues from 10-month-old *hHGFtg:H11^hMET/+^:Ctnnb1^L(Ex3)/+^:PB^Cre4^* mice. **b** Representative images of co-IF staining of SP1 with β-catenin or XPO1 in prostate tissues of *hHGFtg:H11^hMET/+^:Ctnnb1^L(Ex3)/+^:PB^Cre4^* mice. **c** The scheme of the *Xpo1* gene locus and ChIP-qPCR analyses on SP1 binding sites using indicted antibodies. **d** qPCR analysis of *SP1* and *XPO1* shown as fold change in PC3 cells, DU145 cells, and organoids derived from dissected prostate cells of

*hHGFtg:H11^hMET/+^:Ctnnb1^L(Ex3)/+^:PB^Cre4^* mice. **e** Immunoblotting of cell lysates after indicated shRNA treatment showing protein expression of SP1, XPO1, and Actin in PC3, DU145, and prostate organoid cells derived from PCa tumors of *hHGFtg:H11^hMET/+^:Ctnnb1^L(Ex3)/+^:PB^Cre4^* mice. Actin was used as an internal standard. In **c** and **d**, data are represented as mean ± s.d. of three biological replicates. **P < 0.01, ***P < 0.001; Unpaired two-tailed *t*-tests. In **a**, **b**, and **e**, representative images with consistent results from three biological replicates are shown. Scale bars, 25 μm. Source data and the exact *P* values are provided in the Source Data file.

without DHT (Fig. 6c). Measuring average sizes of individual organoids and the organoid forming efficiency further affirmed the inhibitory effects of ENZ, Selinexor, and CX5461 (Fig. 6d, e). The effect of the above inhibitors on PCa growth was further examined using in vivo tissue grafting assays (Fig. 6a). The size and weight of prostatic tumor grafts treated with ENZ were significantly smaller and less than vehicle-treated counterparts in intact hosts, but no different in castrated mice (Fig. 6f, g), confirming the androgen-independent growth capacity of PCa cells developed from TripleTg mice. Grafts treated with Selinexor were significantly smaller and weighed less than vehicle-treated samples in both intact and castrated hosts, and CX5461 showed a greater inhibitory effect in castrated hosts than in intact hosts (Fig. 6f, g). Histological analyses showed less differentiated tumor characteristics in vehicle-treated grafts from castrated host in comparison to those from intact hosts (Fig. 6h). Pathological changes similar to ADT-induced tumor regression were exhibited in ENZ-treated grafts from intact hosts, and no tumor lesions appeared in Selinexor or CX5461-treated grafted samples (Fig. 6h). Measuring the number of Ki67+ cells in the above samples correlated with the effects of the different

inhibitors as shown in both gross and pathological analyses (Fig. 6g). Taken together, the above data reaffirm the promotional role of XPO1 and ribosomal biogenesis activation in androgen-independent PCa growth, providing additional experimental evidence for targeting these oncogenic pathways to prevent PCa progression, and DNPC development.

**Aberrant activation of HGF/MET and Wnt axes promotes mCRPC development**

Given current ADT directly induces DNPC development, we examined the effect of co-activation of HGF/MET and Wnt/β-catenin in ligand-independent PCa growth, progression, and mCRPC development. TripleTg mice were castrated at 4 months and analyzed at 8 months of age (Fig. 7a). Multiple invasive tumor lesions occurred both locally and at the distant sites in castrated mice compared to age- and genotype-matched intact counterparts (Fig. 7a, b). Histological analyses identified poorly differentiated tumor lesions in both primary and metastatic tumor samples, sharing very similar cellular characteristics as observed in Solid-PCa cells (Fig. 7c). Specific expression of pMET and

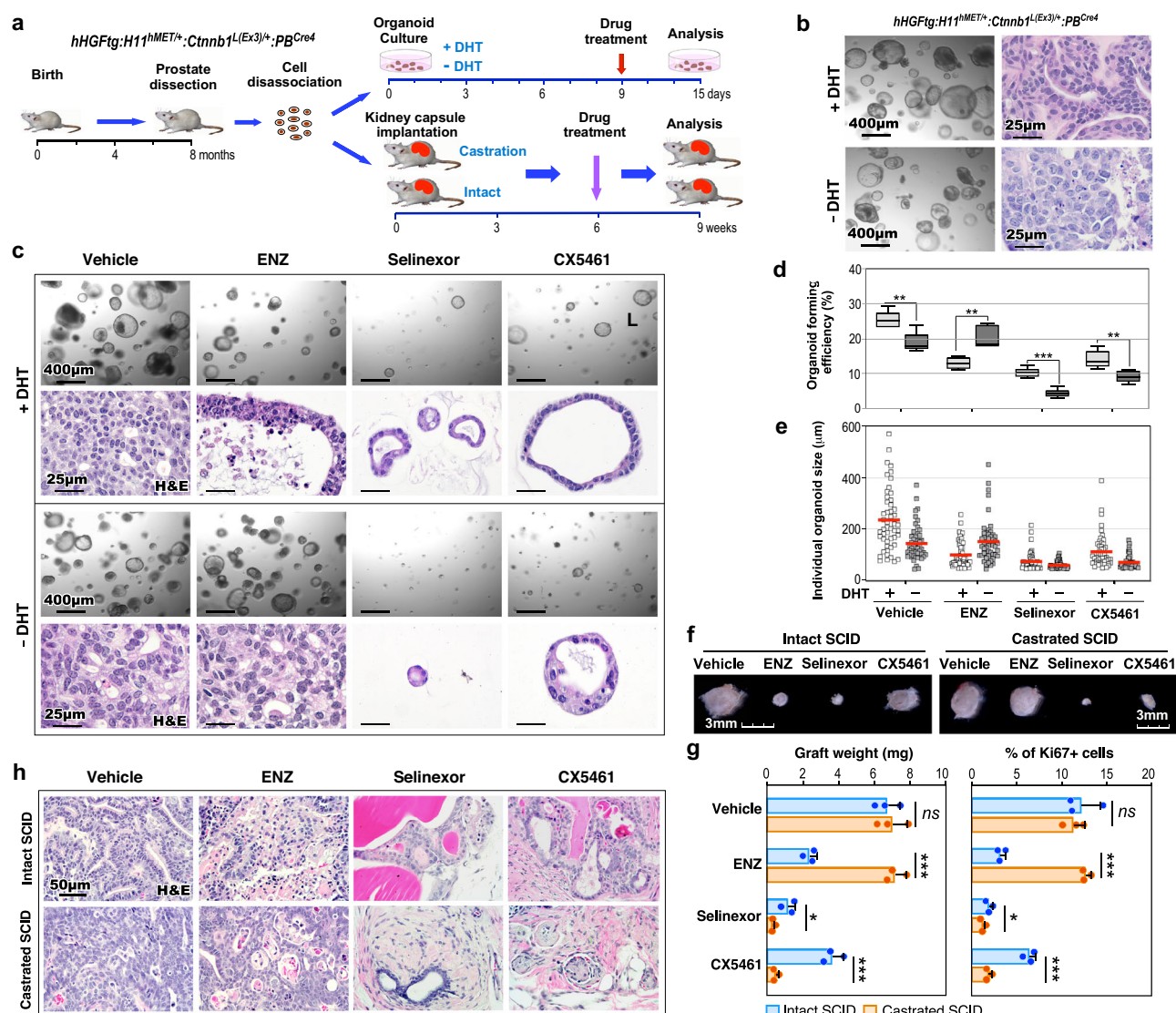

**Fig. 6 | Aberrant activation of XPO1 and ribosomal synthesis pathways converts PCa cellular properties and promotes androgen-independent growth.**
**a** Schematic representation of the experimental design for the organoid culture and kidney capsule transplantation assays. Organoids derived from dissected prostate cells of *hHGFtg:H11^hMET/+^:Ctnnb1^L(Ex3)/+^:PB^Cre4^* mice were developed and then treated with vehicle and different inhibitors in the presence or absence of DHT for 6 days. Intact and castrated SCID host mice transplanted with dissected prostate cells of *hHGFtg:H11^hMET/+^:Ctnnb1^L(Ex3)/+^:PB^Cre4^* mice were administrated with vehicle and different inhibitors. See "Methods" section. **b** Representative images of brightfield and H&E staining for organoids in the presence or absence of DHT. **c** Representative images of brightfield and H&E staining for organoids with indicated treatments. **d** Quantification of organoid forming efficiency showing the percentage of organoids above 50 μm diameter per total cells seeded at day 0 in a well. The center line represents the median value, the box borders represent the lower and upper quartiles (25% and 75% percentiles, respectively), and the ends of the bottom and top whiskers represent the minimum and maximum values, respectively, for five independent samples over three biological replicates. **e** Quantification of individual organoid size. Organoids per treatment group (*n* = 50) examined over three independent experiments. The center red bar indicates the mean value in each group. **f**–**h** Representative image for gross (**f**) or H&E staining (**h**) of xenografts with the indicated treatments. Weights of xenografts (*n* = 3; left) and quantification of Ki67+ cells per total cells (right) from groups treated as indicated (**g**). Data are represented as mean ± s.d. of three biological replicates. Representative images from three biological replicates are shown. In **d** and **g**, *P* < 0.05, **P* < 0.01, ***P* < 0.001; Unpaired two-tailed *t*-tests. Source data and the exact *P* values are provided in the Source Data file.

nuclear β-catenin with a lack of nuclear AR and SYN expression was observed in both prostate and lung metastatic tumor cells (Fig. 7c), confirming the double-null cell properties of mCRPC in castrated TripleTg mice. To gain in-depth insight into transcriptomic changes induced by castration, we analyzed bulk RNA-sequencing (RNA-seq) samples prepared from microscopically confirmed PCa tissues of both intact and castrated TripleTg mice (Fig. 7d). GSEA using pre-ranked gene lists from the above DEGs between castrated and intact samples revealed significant enrichment in pathways related to PCa oncogenesis, including HGF, E2F, EMT, β-catenin, and KRAS activation as well as tumor invasion and progression (Fig. 7e). Results from qRT-PCR

analyses showed higher expression of *Myc*, *Xpo1*, *Rpl12*, *Rps16*, and *Eif4a1* in both intact and castrated PCa samples from TripleTg mice than those from DoubleTg mice. A significant increase in *Xpo1* expression was observed in castrated versus intact samples from TripleTg mice (Fig. 7f), reaffirming aberrant activation of XPO1 in PCa progression and DNPC development. Positive staining for XPO1, RPL12, RPS16, and pS6 proteins was also observed in both primary prostate and lung metastatic tumor cells developed in castrated TripleTg mice (Supplementary Fig. 6a). Using prostatic organoid culture organoids, we further assessed the inhibition of XPO1 and ribosomal biogenesis pathways in CRPC cells from TripleTg mice (Fig. 7g). Similar size and

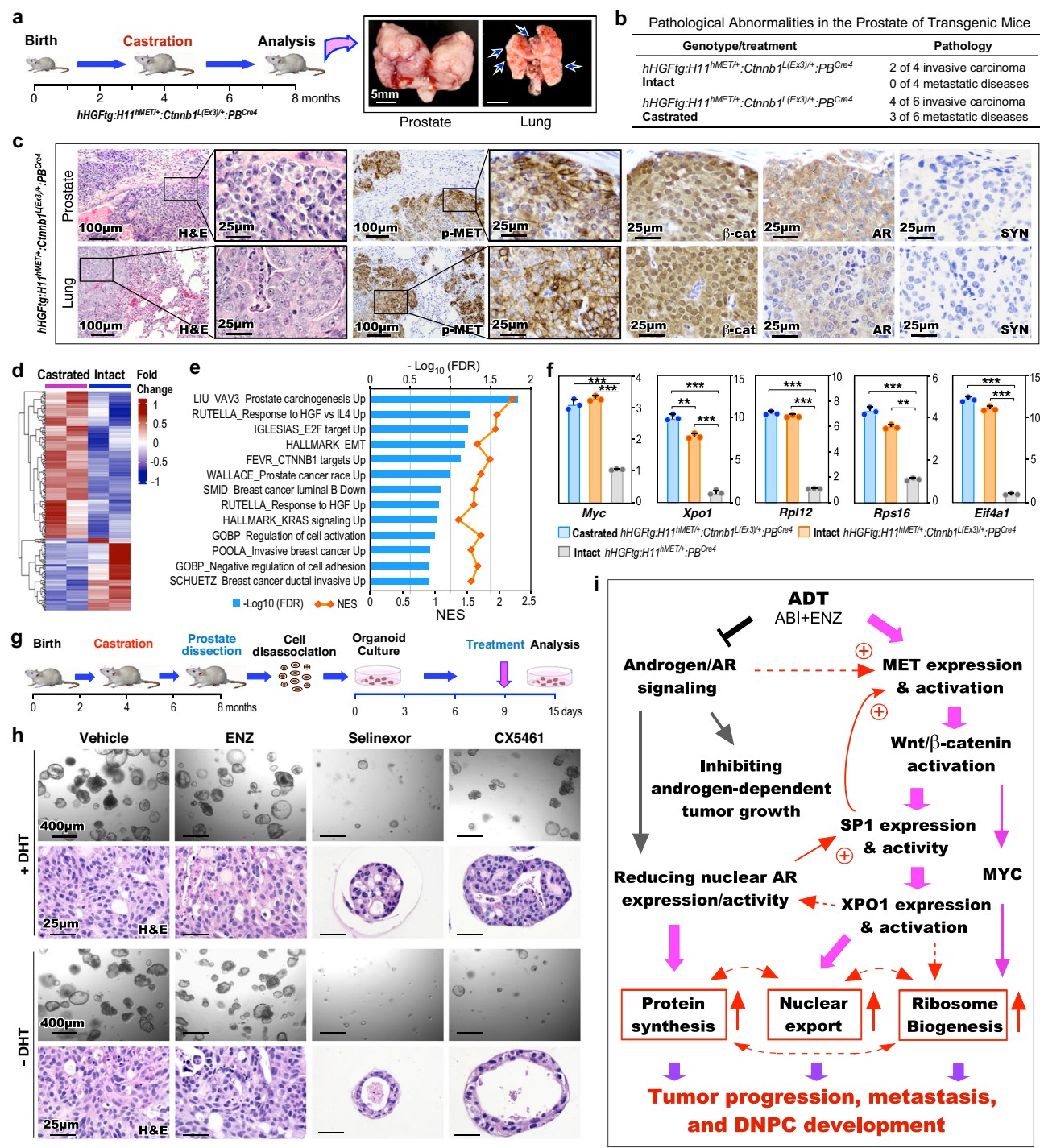

**Fig. 7 | XPO1 activation mediated by HGF/Wnt pathways promotes androgen-independent DNPC progression. a** Schematic representation of experimental design and representative gross images of primary prostate tumors and lung metastases from castrated *hHGFtg:H11^{hMET/+}:Ctnnb1^{L(Ex3)/+}:PB^{Cre4}* mice. **b** Table summarizing the pathological abnormalities in the prostates of intact and castrated mice. **c** Representative images of H&E and IHC staining using the indicated antibodies on adjacent prostate and lung tissues **d** Heatmap showing the expression patterns of DEGs from the comparisons of bulk RNA-seq data from prostate tissues from castrated and intact *hHGFtg:H11^{hMET/+}:Ctnnb1^{L(Ex3)/+}:PB^{Cre4}* mice. Red and blue colors indicate up- and down-regulation, respectively. See Supplementary Data 7. **e** GSEA results from pre-ranked DEG list comparing castrated versus intact TripleTg samples. **f** qPCR analysis of the indicated genes shown as fold change in prostate

tissues from the indicated mice. Data are mean ± s.d. of three biological replicates. ***P* < 0.01, ****P* < 0.001; Unpaired two-tailed *t*-tests. **g** Schematic representation of the experimental design for the ex vivo organoid culture performed. Organoids derived from prostate tumor cells of castrated *hHGFtg:H11^{hMET/+}:Ctnnb1^{L(Ex3)/+}:PB^{Cre4}* mice were treated with vehicle, ENZ, Selinexor, or CX5461 in presence or absence of DHT for 6 days. See Methods section. **h** Representative images of brightfield and H&E staining of the organoids with the indicated treatments. **i** Schematic of hypothetic models by which current ADT induces DNPC development through HGF/Wnt-induced activation of XPO1 and ribosome biogenesis through SP1. Representative images with consistent results from three biological replicates are shown. Scale bars, 5 mm (**a**), 400 μm (**h**), 100 μm (**c**), 25 μm (**c**, **h**). Source data and the exact *P* values are provided in the Source Data file.

number of organoids developed in culture conditions either with or without DHT, and vehicle- and ENZ-treated organoids cultured with or without DHT showed no significant difference (Fig. 7h). However, Selinexor- or CX5461-treated organoids appeared significantly decreased in number and size compared to vehicle-treated samples, while the former appeared more potent than the later (Fig. 7h). Quantifying average sizes of organoids, their forming efficiency, and percentages of Ki67+ cells in organoids showed the significant inhibitory effect in Selinexor- or CX5461-treated samples in comparison to vehicle-treated counterparts (Supplementary Fig. 7b–d). These data further confirm the effect of XPO1 and ribosomal synthesis pathway inhibitors on CRPC growth, implicating them as effective and potential targets for treating DNPC.

## Discussion

Current ADT using the second-generation androgen blockers and AR inhibitors, such as ABI and ENZ, yielded more diversity amongst CRPC subtypes, including DNPC[10,35]. The molecular mechanisms for DNPC development are still largely unknown. Additionally, lacking the biologically relevant GEMMs and other in vivo models to investigate DNPC pathogenesis is another challenge in the field. In this study, we directly address these important and urgent questions. Analyzing human CRPC samples identified significantly higher expression of HGF/MET signaling downstream targets in DNPC versus ARPC subtypes[10,13]. A significant correlation between upregulated HGF/MET downstream targets and down-regulated AR downstream targets was further observed in DNPC cells. Additionally, upregulated HGF/MET signaling was also correlated with activation of the Wnt/β-catenin axis in those DNPC samples. Since it has been shown that aberrant activation of HGF/MET and Wnt/β-catenin pathways can be regulated by ADT and closely associated with PCa progression and poor clinical prognosis[36–39], our findings provide mechanistic insight into ADT-induced PCa progression and DNPC development. In this study, we investigated the pathogenesis of DNPC through aberrant activation of HGF/MET and Wnt/β-catenin pathways using relevant GEMMs, scRNA-seq, and other relevant experimental analyses and approaches.

To assess the effect of HGF/MET aberrant activation in prostate tumorigenesis, we developed human HGF and MET dual transgenic mice (DoubleTg), in which we mimicked the paracrine interaction between HGF derived from prostate stroma and elevated MET expression in prostate epithelia as observed in PCa patients[15–17]. DoubleTg mice showed elevated HGF-induced reciprocal MET activation in the prostate epithelium, leading to prostate oncogenic transformation and PIN and PCa development. Using scRNA-seq and other approaches, we identified a regulatory loop through reciprocal activated HGF/MET signaling to elevate Wnt/β-catenin axes in mouse PCa cells. These findings corroborate our observations of the correlation between the activation of HGF/MET and Wnt/β-catenin signaling pathways in human CRPC samples and implicate a regulatory mechanism for co-activation of these pathways in promoting tumor progression and CRPC development. To directly test this hypothesis, we developed the TripleTg mouse model, in which both the HGF/MET and canonical Wnt signaling pathways are concomitantly activated (Fig. 3a). The TripleTg mice developed the fast-growing and invasive PCa lesions with both local invasion and distant metastases. Progressive pathological lesions corresponding to robust aggressive tumor behaviors were identified in both primary and metastatic tumor lesions. Specifically, a subpopulation of poorly differentiated tumor cells with the cellular properties of lacking both nuclear AR and SYN expression, termed Solid-PCa, was identified in both primary and metastatic PCa lesions. Solid-PCa cells were sustainable through the course of castration and retained their growth ability in the absence of androgens in both in vivo and ex vivo systems. Single-cell trajectory analyses further demonstrated that reduced AR downstream activation with elevated HGF/MET and Wnt/β-catenin signaling pathways

simultaneously occurred in these tumor cells through PCa progression, providing a molecular basis for their fast-growing and aggressive cellular properties. These lines of experimental evidence demonstrate the critical role of HGF/MET and Wnt/β-catenin signaling co-activation in promoting tumor progression and inducing AR and SYN double-negative tumor cells, which is similar to what we have observed in PCa patient samples treated with current ADT.

To understand the regulatory role of HGF/MET and Wnt/β-catenin signaling pathways in promoting DNPC-like tumor development, we assessed the transcriptomics of PCa cells from TripleTg mice. A significant increase in the expression of XPO1/CRM1, ribosomal proteins, and translational initiation factor, EIF4A1, was identified in poorly differentiated Solid-PCa cells. Additionally, a promotional role for stabilized β-catenin in augmenting SP1-regulated XPO1 expression was observed in these PCa cells. Increased expression of XPO1 and ribosomal proteins was further demonstrated in human CRPC samples from ABI- and ENZ-treated patients. Analyses of human scRNA-seq and bulk RNA-seq datasets also showed a significant increase in the expression of *XPO1* in CRPC cells that also possess activated Wnt/β-catenin signaling pathways[14], directly supporting the clinical relevance of aberrant activation of Wnt/β-catenin in increasing XPO1 expression during PCa progression. Emerging evidence has shown a critical role of XPO1 in regulating the nuclear export of proteins and RNA molecules[32,33]. Aberrant activation and alteration of XPO1 directly regulate tumor progression, metastasis, and drug resistance in various human malignancies[32–34]. Selinexor, a FDA-approved XPO1 inhibitor, has shown antitumor activity in the clinic and is currently being used to treat various human tumors, including melanoma, colon cancer, glioblastoma, and small cell lung cancer[40–43]. Treating mCRPC patients with Selinexor has shown visible clinical activity, unfortunately also with poor tolerability[11]. Identifying increased XPO1 expression in Solid-PCa cells of TripleTg mice and human CRPC samples in this study provided additional experimental evidence for the oncogenic role of XPO1 in advancing PCa progression, hormone refractoriness, and CRPC development, which further supports the development of the relevant therapeutic strategies for treating advanced PCa. In this study, we re-evaluated the effect of Selinexor using both PCa organoids and grafts derived from TripleTg mice. A significant inhibitory effect of Selinexor on AR-dependent and -independent PCa tumor growth was observed both in vivo and ex vivo, suggesting that XPO1 blockade can repress both AR and other oncogenic pathway-mediated PCa growth. Whereas these data need to be validated in relevant human samples and cell lines, they suggest a potential combinational therapeutic strategy with ADT for future treatment of advanced PCa. Additionally, observations of increased XPO1 expression in castrated TripleTg mouse PCa samples and clinical samples of ABI- and ENZ-treated patients further support the promotional role of XPO1 in PCa progression, hormone refractoriness, and DNPC development. Therefore, more studies using biologically relevant human DNPC samples and cell lines warrant further validation our data from the above GEMMs to fully understand the pathogenesis of DNPC.

Increases in nucleolar size and number, resulting from abnormal ribosomal synthesis, have been considered as hallmarks for fast-growing and aggressive tumor cells and are closely correlated with poor prognosis[22]. Similar nucleolar abnormalities were also identified in PCa cells, particularly in Solid-PCa cells of TripleTg mice. Elevated expression of MYC, a master regulator of ribosome biogenesis[28], has been observed in Solid-PCa cells (Fig. 4k). Accordingly, elevated expression of RPs and *Eif4a1* was also detected in both prostate primary and metastatic tumor cells, aligning with upregulated XPO1, HGF/MET, and Wnt/β-catenin downstream target expression. In this study, we also observed the effect of XPO1 inhibition on reducing both global translation and decreasing nucleolar size and number in PCa cells derived from TripleTg mice (Supplementary Fig. 7a–d). These data suggest the important role of XPO1 in facilitating rRNAs, RPs, and

assembly factors in ribosomal biogenesis and protein synthesis during tumor growth and progression[44]. High-level ribosomal protein amplifications occur frequently in metastatic breast cancer and CRPC samples[45,46], and dual RP inhibitors have shown an inhibitory effect on the growth of patient-derived xenograft (PDX) from CRPC patients[46]. Additionally, reduced AR content increases the assembly of the EIF4F translation initiation complex, thereby promoting cell proliferation[47]. Taken together, these data uncover a regulatory loop initiated by ADT-induced HGF/MET and Wnt/β-catenin signaling activation through dysregulation of XPO1-regulated nuclear export, ribosomal biogenesis, and protein synthesis to promote tumor progression, hormone refractoriness, and DNPC development (Fig. 7i), providing scientific evidence for targeting nuclear export and ribosomal synthesis in combination with current ADT for future therapies.

In TripleTg mice, nuclear AR expression was detected in well-differentiated Adeno-PCa cells with reduced or absent nuclear AR expression in poorly differentiated and invasive Solid-PCa cells. Increased XPO1 expression was also observed in Solid-PCa cells in comparison to Adeno-PCa cells. Our confocal microscopic analyses showed more intense peri-nuclear staining of XPO1 corresponding to less AR nuclear staining in Solid-PCa than Adeno-PCa cells of TripleTg samples (Supplementary Fig. 7e). An increase in the expression of Nups, the components of NPCs, was also observed in Solid-PCa cells, co-localizing with XPO1 (Supplementary Fig. 7e). These data suggest a potential role for aberrant activation of XPO1 interacting with NPC components in regulating AR nuclear exporting in Solid-PCa cells. It has been shown that there is an interaction between XPO1 and AR through GSK3β signaling pathways[48]. However, XPO1 inhibition on PCa cells derived from TripleTg mice failed to show an increase in nuclear AR expression in organoid cultures. Moreover, lack of AR nuclear expression was more prominent in both primary and metastatic tumor lesions from castrated TripleTg mice. Given that current ADT directly induces DNPC development[10], the above observations suggest that the lack of AR nuclear expression observed in mouse "Solid-PCa" cells may be regulated through aberrant nuclear import and export pathways. Additionally, a ligand-independent nuclear import mechanism for AR has also been observed in PCa cells[49]. In the future, more investigations should be performed using biologically relevant human DNPC samples and cell lines to validate these data from the GEMMs. Data generated from these studies will provide fresh insight on the regulatory mechanisms underlying DNPC development and lead to the development of more effective therapies for advanced PCa.

## Methods

### Ethics Statement

All experimental procedures and care of animals in this study were carried out according to the Institutional Animal Care and Use Committee (IACUC) at Beckman Research Institute of City of Hope (California, US) and approved by the IACUC. Euthanasia was performed by $CO_2$ inhalation followed by cervical dislocation. Mice housing conditions are under a 12 h light/dark cycle at 20–24 °C and 30–70% humidity in our institution.

The study of human prostate cancer tissues was approved by the Institutional Review Board (IRB)-approved protocol (IRB # HS-16-00817) at the University of Southern California, and informed Consent was obtained from all participants in this study. Given the nature of prostate cancer, we only used male human and mouse samples in this study.

### Human samples and data analysis

Human prostate cancer samples were used in this study. They included five specimens of prostatectomies from 5 patients without hormonal treatment, and the metastatic CRPC specimens from 6 individual patients who had previously received second-line antiandrogen therapies- ENZ and ABI- for at least 3 months after the failure of first-line ADT treatment, including two samples from lymph node, three from brain, and one from adrenal gland. These tumor samples were of high tumor purity. RNA-seq data for metastatic prostate adenocarcinoma patients (total 266 polyA RNA samples from SU2C/PCF) and drug treatment status per sample were downloaded from the cBioPortal website (https://www.cbioportal.org/study/summary?id=prad_su2c_2019)[50]. NE and AR signaling scores for each sample were computed using the $\log_2$-transformed FPKM values as input and 10 NE-associated genes and 10 AR-associated genes as previously described[10,13]. These mCRPC samples were classified as ARPC (AR+/NE−), NEPC (AR−/NE+), DNPC (AR−/NE−), AR+/NElow, ARlow/NE+, and AR+/NE+[13]. Tumor samples classified as ARPC, NEPC, and DNPC samples were further analyzed in this study. Additionally, RNA transcripts per million (TPM) data for mCRPC patients (total 210 samples from the West Coast Dream Team) were downloaded from https://quigleylab.ucsf.edu/data. Expression-based subtypes of mCRPC were identified as ARPC (AR+/NE−), ALNN (ARlow/NE−), ARNP (AR+/NE+), NEPC (AR−/NE−), and DNPC (AR−/NE−) using hierarchical clustering, as previously described[29]. DEGs were considered using Wilcoxon Rank Sum tests and a value of |$\log_2$ fold change| > 0.1 and adjusted P-values < 0.05. P values were adjusted for multiple testing with Benjamini−Hochberg correction. Pathway analysis was performed using Gene Set Enrichment Analysis (GSEA 4.3.2). Groupwise co-expression analysis was performed using a log-rank test to identify the significance of the Spearman's correlation coefficient between the mRNA expression z-scores using the hacksig R package[51] in R. We obtained scRNA-seq datasets from both human naïve primary PCa and ADT-treated mCRPC tissues, which have been deposited in NCBI's SRA database and is accessible through SRA accession: PRJNA699369.

### Mouse generation, mating, and genotyping

The human *HGF* transgenic (*hHGFtg*) mice were purchased from The Jackson Laboratory (strain #: 030205)[52]. To generate the human *MET* transgenic (*H11^hMET^*) mouse line, we used integrase-mediated transgenesis technology as shown before[53]. In brief, human *MET* cDNA was sub-cloned into the pB378 vector containing an *attB* recombination site. A *loxP-PGK-Neomycin- STOP-loxP* cassette was inserted between the CAG promoter and a flag-tagged human *MET* coding sequence followed by a polyadenylation signal (Fig. 2a), in which the *hMETtg* expression can be achieved in a constitutive but tissue-specific manner through *Cre* recombinase-mediated removal of the *LSL* cassette. The DNA was purified and microinjected along with φC31 integrase mRNA into zygotic pro-nuclei of an FVB mouse that contains *attP* docking site at *H11* locus. Correctly targeted mice were screened by mouse tail tissue genomic PCR using P1 (5′-TGACCAGTGGGACTGCTTTTT-3′) and P3 (5′-CACACGACCAGGCCTTCCTTCTT-3′) primers, and further confirmed by DNA sequencing. PB-Cre4 (*PB^Cre4^*) mice were obtained from the NCI mouse repository (strain #: 01XF5). Mice containing the conditional *Ctnnb1* allele (*Ctnnb1^L(Ex3)^*) were kindly gifted from Dr. Makoto M. Taketo[54]. *Human HGFtg:H11^hMET/+^* female mice were first generated by intercrossing *hHGFtg* males with *H11^hMET/hMET^* female mice and then mated with *PB^Cre4^* males to generate *H11^hMET/+^:PB^Cre4^* and *hHGFtg:H11^hMET/+^:PB^Cre4^* littermates at age of 3, 6, and 10 months (Fig. 2a). *Ctnnb1^L(Ex3)/+^:PB^Cre4^* were generated by intercrossing *Ctnnb1^L(Ex3)/L(Ex3)^* females with *PB^Cre4^* male mice. Similar mating procedures were used to generate *hHGFtg:H11^hMET/+^:Ctnnb1^L(Ex3)/+^:PB^Cre4^* mice at age of 3, 6, and 10 months (Fig. 3a). Experimental mice generated in this study were mixed from C57BL/6 and C3H SCID backgrounds. Genomic DNA samples isolated from mouse tail tips were used for genotyping with appropriate primers (Supplementary Table 1). All mice-bearing tumors were closely monitored during the entire course of the study, and maximal tumor sizes (1.5 cm in diameter) were not exceeded based on the guidelines of IACUC at Beckman Research Institute of City of Hope.

## Serum HGF measurement

HGF levels in mouse sera were measured using a human HGF Quantikine ELISA kit (DHG00B, R&D Systems). The samples were collected from ten biologically different mice, measured in duplicate, tested, and calculated following the manufacturer's protocol.

## Mouse experiments

Adult male mice were castrated. For in vivo kidney capsule implantation, cells were isolated and dissected from the mouse prostate tissues at the indicated timepoint. Once trimmed, prostate tissues were cut into small pieces and digested in Gibco™ Dulbecco's Modified Eagle's Medium (DMEM)/Ham's F12 50/50 Mix (DMEM/F12) (11320033, Fisher Scientific) supplemented with 10% Gibco™ Fetal Bovine Serum, 1 nM dihydrotestosterone (DHT) (A8380, Sigma Aldrich), 10 μM Y-27632 dihydrochloride (M1817, Abmole Bioscience), and 1 mg/ml type II collagenase (17101-015, Life Technologies) at 37 °C for 135 min with rotation at 150 rpm, and then in Gibco™ TrypLE (12605-028, Fisher Scientific) supplemented with 1 nM DHT, 10 μM Y-27632 dihydrochloride and 0.5 U/μl DNase I (D5025, Sigma Aldrich) at 37 °C for 30 min with rotation. Cells were filtered through a 37 μm cell strainer (27215, StemCell Technologies) and resuspended in PBS with 0.05% Bovine Serum Albumin. Cell viability and number were detected using a TC Automated Cell Counter (Bio-Rad Laboratories), and only cells with at least 80% viability were processed. Approximately $1 \times 10^5$ PCa cells were transplanted under the renal capsule of either intact or castrated 6–8-week-old male NOD/SCID mice. 6 weeks after implantation, mice were randomized into vehicle control and treatment groups and were orally administered with vehicle (10% DMSO in corn oil), Enzalutamide[55], Selinexor[56], or CX5461[57] at 20, 10, or 30 mg/kg body weight, respectively. The compounds were administered three times per week for 21 days. Xenografts were excised, weighed, and fixed for histological analysis.

## Histological analyses, AgNOR staining, and immunostaining

Mouse tissues were fixed in 10% neutral-buffered formalin (American Master Tech Scientific) and processed into paraffin. Following paraffin embedding, tissue blocks were cut to 4 μm serial sections. For histological assessment, hematoxylin and eosin (H&E) staining was performed[58]. Pathological analyses were conducted in accordance with the guidelines suggested by The Mouse Models of Human Cancers Consortium Prostate Pathology Committee in 2013[18]. For AgNOR staining, unstained sections were deparaffinized, rehydrated through a decreasing ethanol gradient, and boiled in 0.01 M citrate buffer (pH 6.0). The sections were pre-treated with 0.3% acetic acid and then incubated with silver nitrate solution in a dark humidified chamber for 13 min at 37 °C, followed by dehydration through an increasing ethanol gradient, and coverslips were mounted with Permount Medium (Thermo Fisher Scientific). The silver staining solution consisted of 33% silver nitrate and 0.6% gelatin in 0.33% formic acid solution. For immunohistochemistry (IHC), tissue sections were rehydrated through a decreasing ethanol gradient, treated by boiling in 0.01 M citrate buffer (pH 6.0) or Tris-EDTA (pH 9.0) for antigen retrieval, incubated in 0.3% $H_2O_2$ for 15 min, blocked by 5% normal donkey serum (Gibco) in phosphate-buffered saline (PBS) for 1 h at room temperature, and incubated with indicated primary antibody diluted in 1% normal donkey serum at 4 °C overnight. Next, the slides were then incubated with biotinylated secondary antibody for 1 h followed by horseradish peroxidase streptavidin (Vector Laboratories) for 30 min and visualized using a DAB kit (Vector Laboratories). After counterstaining was performed with 5% (w/v) Harris Hematoxylin (Thermo Fisher Scientific), slides were dehydrated, and cover slipped. For IF staining, tissues sections underwent heat-induced epitope retrieval, blocked in 5% normal donkey serum, and incubated with primary antibodies at 4 °C overnight and with fluorescent-conjugated secondary antibodies, and then mounted with Vectashield Mounting Medium with DAPI (H-1200-10, Vector Laboratories). All antibodies used in this study are detailed in Supplementary Table 2.

## Transmission Electron Microscopy (TEM)

For nuclear pore complex and cellular morphological analyses, prostate tissues were fixed with 4% paraformaldehyde with 2.5% glutaraldehyde, 0.2 M sodium cacodylate buffered at pH 7.4, and 2 mM $CaCl_2$. Standard tissue sample preparation for TEM was followed including post-fixation with osmium tetroxide, serial dehydration with ethanol, and embedment in Eponate. Electron microscopy (EM) images were collected with an FEI Tecnai 12 transmission electron microscope (Thermo Fisher Scientific) equipped with a LaB6 filament and operated at an acceleration voltage of 120 kV. Images were taken with a Gatan OneView CMOS camera (Gatan). Number of nuclear pore complex per nuclear membrane length and nuclear size were quantified with the Image J (NIH) using seven different areas from three biologically different samples per group.

## Single-cell suspension preparation and single-cell RNA-sequencing analysis

Two individual sets of scRNA-seq experiments were performed using different littermates in this study. Prostate tissues were isolated at indicated timepoints and placed in DMEM/F12 with 10% FBS, 10 nM DHT, and 10 μM Y-27632 dihydrochloride, and digested using 1 mg/ml type II collagenase at 37 °C for 90 min and TrypLE (Gibco) supplemented with 10 nM DHT, 10 μM Y-27632 dihydrochloride and 0.5 U/μl DNase I at 37 °C for 15 min. Then single cells were loaded on the Chromium Controller (10× Genomics) targeting 8000–12,000 cells per sample. Single-cell RNA-seq libraries were prepared using 10× Genomics Chromium Single Cell 3′ Solution with v3 chemistry according to the manufacturer's protocol. The purity and size of cDNA were validated by capillary electrophoresis on Agilent 2100 Bioanalyzer using Agilent High Sensitivity DNA Kit (#5067-4626, Agilent Technologies). The library quantity was measured using Qubit dsDNA HS Assay Kit (Invitrogen). cDNA libraries were sequenced on Illumina Novaseq 6000 S4 flow cell (Illumina) to a depth of 50–100 K reads per cell. Processing of raw sequencing data, including FASTQ file generation and Unique Molecular Identifiers (UMI) counting, was conducted using the 10× Genomics Cell Ranger pipeline (6.0.2). Reads were aligned to the mm10 genome with *hHGFtg* and *H11^hMET^* sequences for gene expression count. The Seurat package (4.3.0) in R (4.2.2) was used for the subsequent data analysis following upload of a filtered feature bar-coded matrix including 12,004 and 14,939 cells from *hHGFtg:H11^hMET/+^:PB^Cre4^* and *hHGFtg:H11^hMET/+^:Ctnnb1^L(Ex3)/+^:PB^Cre4^* prostate tissues, respectively. For a quality-control step, cells then underwent further filtering to remove potential empty droplets and doublets (200 < nFeature_RNA < 7000) as well as low-quality cells with high percentages of mitochondrial RNA (percent.mt < 15). Following this final filtering, 9236 *hHGFtg:H11^hMET/+^:PB^Cre4^* cells with an average of 3822 genes per cell and 27,456 UMI counts per cell, and 9492 *hHGFtg:H11^hMET/+^:Ctnnb1^L(Ex3)/+^:PB^Cre4^* cells with an average of 3500 genes per cell and 24,195 UMI counts per cell were conserved for future analyses. Integrating datasets were performed on Seurat using *FindIntegrationAnchors* and *IntegrateData* functions with twenty dimensions. Data was visualized by a nonlinear dimensionality reduction UMAP technique using the Seurat *RunUMAP* function.

For analysis of signaling pathways, differential expression analysis was performed on Seurat using *FindMarkers* or *FindAllMarkers* functions. *P* values were adjusted for multiple testing with Benjamini–Hochberg correction. Genes with |average log$_2$ fold change| > 0.1 and adjusted *P*-value < 0.05 were considered as differentially expressed genes (DEGs). We used Gene Set Enrichment Analysis (GSEA) (4.3.2) analysis with hallmark, gene ontology, and curated gene sets. Spearman pairwise correlation matrices as the measure of association between *hMETtg* and other genes were analyzed and

 

plotted using the *ggcorr* function from the GGally R package (https://www.rdocumentation.org/packages/GGally/versions/1.5.0/topics/ggcorr) in R.

For pseudotime generation and trajectory analysis, Seurat objects were converted into CellDataSet format, and the cell trajectory was calculated on the Monocle3 package (0.2.0)[59] using *learn_graph* function. To order the cells according to pseudotime, the BE clusters were selected as the root nodes. The aligned kinetic curves showing gene expression along pseudotime were generated using *plot_genes_in_pseudotime* function.

## RNA extraction and qRT-PCR
Total RNA was extracted from fresh mouse prostate tissues, lung tissues, or prostatic organoids derived from prostate tissues of WT, *hHGFtg:H11^{hMET/+}:PB^{Cre4}*, or *hHGFtg:H11^{hMET/+}:Ctnnb1^{L(Ex3)/+}:PB^{Cre4}* mice and human prostate cancer cells using RNeasy Mini Kit (#74104, Qiagen) and reverse-transcribed using SuperScript IV First-Strand Synthesis System (#18091050, Fisher Scientific) according to the manufacturer's protocol. qRT-PCR were conducted on the 7500 Real-Time PCR system (Fisher Scientific) using Power-SYBR Green PCR Master Mix (4367659, Applied Biosystems) with specific primers (Supplementary Table 1). Relative quantification was normalized to the level of mouse peptidylprolyl isomerase A (*Ppia*) or human 18S ribosomal RNA and was calculated using the comparative $C_T$ method[60].

## Chromatin immunoprecipitation (ChIP)-qPCR
ChIP-DNA prostate tissues of *hHGFtg:H11^{hMET/+}:PB^{Cre4}* and *hHGFtg:H11^{hMET/+}:Ctnnb1^{L(Ex3)/+}:PB^{Cre4}* mice was obtained by ChIP assay. Briefly, mouse prostate tissues were minced, cross-linked with 1% formaldehyde for 25 min at room temperature (RT), and quenched with 150 mM glycine for 10 min. Samples were homogenized in cell lysis buffer (140 mM NaCl, 50 mM Tris-HCl [pH 8.0], 1 mM EDTA, 10% Glycerol, 0.5% NP-40, and 0.25% Triton X-100). The chromatin was resuspended in nuclear lysis buffer (0.2% SDS, 10 mM Tris-HCl [pH 8.0], 1 mM EDTA, and 0.5 mM EGTA) and fragmented to an average size of 200–500 bp with a Sonic Dismembrator Model 100 (Thermo Fisher Scientific) at 4 °C. After centrifugation, the cell sonicate was diluted with ChIP dilution buffer (0.01% SDS, 167 mM NaCl, 16.7 mM Tris-HCl [pH 8.1], 1.1% Triton X-100, and 1.2 mM EDTA), pre-cleared using Dynabead Protein G (10003D, Invitrogen) and then was subjected to immunoprecipitation with Dynabead Protein G conjugated with SP1 antibody (Novus, NB600-233) or normal IgG (Cell Signaling) for 4 h at 4 °C. Crosslinks were reversed, and then chromatin DNA fragments were analyzed by qRT-PCR with specific primers (Supplementary Table 1).

## Cell culture and shRNA lentivirus infection
PC3 and DU145, human prostate cancer cell lines from the American Tissue Culture Collection (ATCC, CRL-1435, and HTB-81), were maintained in Roswell Park Memorial Institute (RPMI) 1640 medium supplemented with 5% Gibco™ Fetal Bovine Serum in a humidified 5% $CO_2$ incubator at 37 °C. Both PC3 and DU145 were authenticated by the cell bank using DNA profile (STR) and cytogenetic analysis. Control or SP1 shRNA lentiviral particles were purchased from Santa Cruz Biotechnology (sc-108080 or sc-29487-V). Approximately $1 \times 10^5$ cells were plated in a 12-well plate 24 h before infection. Lentivirus infection was carried out with 20 μl of lentiviral particles ($5 \times 10^6$ infectious units of virus (IFU)/ml) in the presence of 6 mg/ml Polybrene (H9268, Sigma Aldrich), and then selected with puromycin (P8833, Sigma Aldrich) after 48 h.

## Western blotting
Cells were harvested in a buffer containing 150 mM NaCl, 2 mM MgCl2, 50 mM HEPES-KOH (pH 7.4), 1 mM EDTA, 5% glycerol, 1 mM dithiothreitol, 0.5 mM phenylmethylsulfonyl fluoride, 25 mM NaF.

Lysates were incubated on ice for 30 min and centrifuged at 14,000 g for 5 min. Protein concentration was measured by the Pierce™ BCA protein assay kit (23227, Thermo Scientific). Lysates boiled in SDS-sample buffer were resolved in 10% SDS-PAGE and transferred onto a nitrocellulose membrane. Membranes were blocked in PBST with 5% Bovine Serum Albumin (Sigma Aldrich) and probed with the anti-ACTIN antibody (A4700, Sigma Aldrich, 1:1000), anti-SP1 antibody (NB600-233, Novus Biologicals, 1:5000), anti-XPO1 antibody (NB100-79802, Novus Biologicals, 1:5000). Anti-rabbit or mouse IgG conjugated to horseradish peroxidase were used as secondary antibodies at the dilution 1:1000 (170-6515 or 170-6516, Bio-Rad). The blots were detected with an enhanced chemiluminescence (ECL) detection kit (GE Healthcare).

## RNA-seq and data analysis
RNA integrity and quality were evaluated using Agilent RNA 6000 Nano Kit (#5067-1511, Agilent Technologies). cDNA libraries were prepared using Kapa RNA HyperPrep Kit with RiboErase (KR1351, Kapa Biosystems) followed by sequencing on Illumina Hiseq 2500 at our institution. Adapter sequences and other redundant sequences were removed from raw paired-end reads using Trimmomatic (v0.39)[61]. The filtered reads were aligned to the mm10 mouse reference genome using STAR (v2.7.9)[62]. Read counts per gene were detected using HTSeq (v2.0.2)[63] and were further normalized using the trimmed mean of M-value method[64] in edgeR (v2.26.7)[65] to obtain counts per million-mapped reads (CPM) and transcripts per million (TPM). The fold change, *P*-value, and a false discovery rate (FDR) were calculated using quasi-likelihood (QL) F-test in edgeR with default settings using normalized expression values, TPM. Only genes with a CPM > 1 in at least two samples were retained for differential analysis. The *P* values were adjusted for multiple testing with Benjamini–Hochberg correction. DEGs were defined as those having a |log$_2$ fold change|> 1 and adjusted *P*-value < 0.05 and were pre-ranked based on the log$_2$ fold change and adjusted *P*-value. GSEA (4.3.2) using pre-ranked gene lists were performed for annotation, integration, and visualization of database to identify network and biological functions of DEGs[66].

## Organoid culture
Prostatic tumor tissue was isolated from either intact or castrated *hHGFtg:H11^{hMET/+}:Ctnnb1^{L(Ex3)/+}:PB^{Cre4}* mice at 8 months of age and dissociated into single cells by digestion in DMEM/F12/FBS/Collagenase for 2 h and then TrypLE for 30 min at 37 °C. Digested cells were passed through 37 μm cell strainers and resuspended in 1:1 (v/v) PBS:Matrigel (#356231, BD Biosciences). For organoid culture, approximately 2000 cells per well were seeded in 24-well plate and cultured in DMEM/F12 containing 0.25 μM A83-01 (R&D Systems), 1× B27 (Life Technologies), 10 ng/mL EGF (PeproTech), 100 ng/mL N-acetylcysteine (PeproTech), 100 ng/mL Noggin (PeproTech), 2.5 ng/mL R-spondin1 (R&D Systems), and 100 μM Y-27632, as shown before but in the presence or absence of 1 nM DHT. Nine days after culture, organoids were treated with vehicle, 0.1% dimethyl sulfoxide (DMSO), antiandrogen, 10 μM Enzalutamide (ENZ), XPO1 inhibitor, 0.5 μM Selinexor (HY-S7252, Selleckchem), RNA polymerase inhibitor, 1 μM CX5461 (HY-13323, MedChemExpress), two times for 6 days and were fixed in 10% neutral-buffered formalin. Fixed cells were subjected to Histogel (HS-4000-012, Thermo Fisher Scientific) and paraffin embedding for histological analysis. Individual organoid size was quantified with the Image J (NIH) using at least 100 organoids per group. Organoid forming efficiency (%) was determined by quantifying the percentage of organoid structure above 50 μm diameter per total cells seeded at day 0 in a well. Quantification of the percentage of Ki67+ cells per total cells was performed from five different areas in four samples per group. All experiments were replicated with two different mice in triplicate wells.

## Detection of protein synthesis

Approximately $1 \times 10^4$ PC3 and DU145 cells were seeded in an 8-well chamber slide (Thermo Fisher Scientific). Cells were treated with vehicle or 1 μM Selinexor for 24 h and then incubated with O-propargyl-puromycin (OPP) for 30 min at 37 °C. The cells were fixed for 15 min in 3.7% formaldehyde and permeabilized with 0.5% Triton X-100 for 30 min. OPP was detected using Click-iT™ Plus OPP Alexa Fluor™ 488 Protein Synthesis Assay Kit (C10456, Thermo Fisher Scientific) then mounted with Vectashield Mounting Medium with DAPI (H-1200-10, Vector Laboratories).

## Microscope image acquisition

Images of H&E and IHC staining were taken by an Axio Lab 1 microscope using 10×, 20×, and 40× Zeiss A-Plan objectives and were captured using a Canon EOS 1000D camera and AxioVision software (Carl Zeiss). Images of IF staining and organoids were acquired on a Nikon ECLIPSE E800 epi-fluorescence microscope at 10×, 20×, and 40× Nikon Plan Fluor objectives using an QImaging RETGA EXi camera with QCapture software (QImaging). Fluorescence images for nuclear pore complex were collected using Zeiss LSM-700 confocal microscope with Zen 2012 Imaging Software.

## Quantification and statistical analysis

Data are presented as the mean values ± s.d. for the indicated number of independently performed experiments. All data are representative of the results of at least three independent experiments. The significance values between data (*$P < 0.05$, **$P < 0.01$, *** $P < 0.001$) were measured using unpaired two-tailed Student's $t$-test or a Wilcoxon Rank Sum test. Adjusted $P$-values were corrected for multiple testing using the Benjamini–Hochberg's procedure. DEG lists were determined using a Wilcoxon Rank Sum test, with genes showing adjusted $P$-value < 0.05 defined to be significant. Spearman's correlation coefficient > 0.3 and $P$-value < 0.05 was considered to indicate a statistical significance. Enrichment scores (ES) for each gene set in the ranked list of genes were calculated by a running-sum statistic using GSEA[66]. Nominal $P$ values of ES were estimated using an empirical phenotype-based permutation test and corrected for multiple hypothesis testing using FDR. As recommended by the GSEA User Guide, pathways with FDR < 0.25 were considered significant in exploratory GSEA pathway analysis.

## Reporting summary

Further information on research design is available in the Nature Portfolio Reporting Summary linked to this article.

# Data availability

RNA-seq processed data for metastatic prostate adenocarcinoma patients (SU2C/PCF Dream Team, 266 samples; polyA assay samples) and drug treatment status per sample were downloaded from the cBioPortal website (http://www.cbioportal.org/). RNA transcripts per million (TPM) data for mCRPC patients (total 210 samples from the West Coast Dream Team) were downloaded from https://quigleylab.ucsf.edu/data. scRNA-seq datasets from both human naïve primary PCa and ADT-treated mCRPC tissues, which have been deposited in NCBI's SRA database and is accessible through SRA accession: PRJNA699369. Raw data of RNA-seq and single-cell RNA-seq have been publicly deposited in Gene Expression Omnibus database under accession number GSE226556. Source Data for this study are provided with this paper. All relevant data in this study are available within the article, Supplementary information, or Source Data. Source data are provided with this paper.

# Code availability

The bioinformatics analyses were conducted using open-source software, including Cell Ranger (v6.0.2), Seurat (v4.3.0), R (v4.2.2), and GSEA (v4.3.2). R scripts used to process sequencing data are available in "GitHub" repository [https://github.com/wk-kim/HGF-WNT-DNPC-Development][67].

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

## Acknowledgements

This work was supported by NIH grants R01CA070297, R01DK104941, R01CA166894, and R01CA233664.

## Author contributions

W.K.K., A.J.B. and Z.S. conceived the project and designed the experiments. W.K.K., A.J.B., D.L., A.H., C.H.N., T.Y., Y.M.A. and Y.B. generated mouse colonies and performed genotyping. W.K.K., A.J.B., D.L., J.W. and A.H. conducted mouse experiments, sample collection, and staining. J.G. and G-Q X. confirmed staining and pathological analyses. W.K.K., A.J.B. and R.V. performed sequence experiments and analyzed sequencing data. W.K.K., A.J.B., Z.L. and D.L. conducted organoid culture experiments and performed staining and analyses. All authors analyzed and confirmed data. W.K.K., A.J.B., A.W.O., M.K., J.G., G-Q X. and Z.S. wrote the manuscript.

## Competing interests

The authors declare no competing interests.
