## [Peer Review File · Nature Communications]

Androgen Deprivation Induces Double-Null Prostate Cancer via Aberrant Nuclear Export and Ribosomal Biogenesis through HGF and Wnt ActivationREVIEWER COMMENTS

Reviewer #1 (Remarks to the Author):

Title: Androgen Deprivation Induces Double-Null Prostate Cancer via Aberrant Activating Nuclear Exporting and Ribosomal Biogenesis by HGF and WNT Axes

Authors: Blinded

Summary

This manuscript reports a series of studies designed to identify and target molecular drivers of prostate cancers that lack both AR signaling and activity of the neuroendocrine (NE) pathway – termed double null prostate cancer (DNPC). Through mining/analyses of human prostate cancer datasets, the authors identified HGF/MET and WNT signaling as differentially increased in DNPC relative to ARPC. Genetically engineered mouse models (GEMMs) were developed that promote MET pathway signaling via cMET and HGF expression, and Wnt pathway activation via *apc* loss. Mice with MET/HGF activation develop low penetrance prostate cancer and reduced Ar activity, whereas the addition of *apc* loss produces high penetrance prostate cancers with variable phenotypes and metastases. Key genes upregulated in these models include XPO1 and ribosome biogenesis. Inhibitors of these features reduced prostate cancer growth.

Comments.

1. Overall, the manuscript is well written and clear. The methods are generally described with sufficient detail that should allow reproducibility. The references cited are appropriate, though there are key omissions (see points 3 and 4).
2. The novel aspects of the present study center on the development of several GEMMs that induce primary and metastatic prostate cancer. These studies indicate that HGF/MET and WNT signaling promote aspects of prostate cancer. These pathways have previously been shown to promote adverse prostate cancer behavior or directly cause prostate cancer in preclinical models and the present work confirms and further extends these observations with detailed single cell based analyses and comparative gene expression studies that nominate potentially druggable pathways.

3. With respect to cMET - prior studies have reported that AR represses MET expression and AR knockdown or tumors with low AR expression activate/upregulate MET expression and activity. (PMID: 24478054; PMID: 23877345; PMID: 20946682; PMID: 26806347; others). These prior reports reduce the novelty of the present work. These studies do not appear to have been referenced.
4. With respect to WNT signaling – prior studies have induced DNPC and PC metastases via *Apc* loss in mouse models, and demonstrated that WNT pathway antagonists repress PC growth - also reducing the novelty of the present findings. These studies do not appear to have been referenced (example: (PMID: 32217460; PMID 32376773).
5. The human data referenced/analyzed by the authors that associate HGF/MET and WNT with DNPC demonstrate that the findings are not unique to DNPC – these pathways are also active/differentially increased in NEPC as observed in Fig1a – this is not discussed.
6. The transgenic models with hMET + hHGF generate primarily PIN lesions – not carcinomas. Overt carcinoma appears to be quite rare. It is not clear what ‘intracystic’ prostate cancer is? This terminology is not used in assessing primary human prostate cancer. The histopathology should be submitted to expert human pathologists for analysis. Detailed images of these tumor should be provided. Even with aging – 9-12 months only 2/11 mice had some evidence of carcinoma.
7. The Wnt pathway activation by hMET+hHGF appears to be insufficient to robustly induce carcinoma. Genetic manipulation to activate Wnt signaling (via *Ctnnb1* manipulation) – did result in more substantial carcinomas and notably metastases were induced. These tumors exhibited a mixed phenotype termed by the authors as adenocarcinoma and ‘solid’ prostate cancer. ‘Solid’ prostate cancer is not a generally accepted histopathological term. The authors should consult with a human prostate pathology expert to establish appropriate nomenclature and determine relevance to human PC.
8. Page 8 Line 162 – the authors state that ‘lack of nuclear AR expression in poorly differentiated Solid-PCa cells corroborates our observations in human DNPC...’ – it is not clear where human DNPC samples were stained for AR in this present study? Showing the murine DNPC next to the human DNPC histology should be performed in a statistical manner to confirm the correlations.
9. The prostates of ‘TripleTg’ mice are shown to exhibit 2 different phenotypes – the ‘adeno’ and the ‘solid’ – but presumably both types and the cells of these types harbor the same

constellation of transgene events – it would be useful to understand what is driving the diversity in phenotype and gene expression.

10. Page 11 Line 233 – the authors state that XP01 and RPL12 are detected in human DNPC samples from ABI- and ENZ treated patients. While notable, (i) the conclusions and validation are not statistically based; and (ii) AR and AR target genes NE genes are not shown. Further – lacking from these samples are data demonstrating HGF/MET or WNT activity in these DNPC tumors – in comparison to ARPC and NEPC. Showing such associations would greatly strengthen the conclusions as clinically relevant.

11. Similarly – the analyzed human RNAseq data showing XP01 expression in two mCRPC epithelial clusters lacks any description/association with AR activity/DNPC phenotypes or in the context of WNT or HGF/MET activity (Expanded Data Fig 5b).

12. Page 15 line 333 – the authors suggest that XP01 is a new and potential target for treating DNPC – first, XP01 is not a new target for prostate cancer overall, and has been evaluated in the context of co-targeting AR in clinical studies where: a) it had very limited effects; and b) there was poor tolerance (PMID: 29487219). Targeting cMet has also had very limited efficacy (PMID: 32943461). This reduces the novelty and potential clinical impact of the findings.

13. The mechanism of Selinexor and XP01 in the context of the model proposed is not clear. If AR loss is mediated by XP01 (nuclear export)– then does XP01 inhibition restore nuclear AR? If AR is lost by some other mechanism – then what is Selinexor inhibition of XP01 doing? Clearly, other transcription factors may be inhibited by XP01 blockade – this is not specific to AR.

14. Page 15 Line 338 – the authors state that there are no biologically relevant in vivo models to investigate DNPC pathogenesis. This is not correct. See PMID 32376773. There are also PDX lines that are DNPC reported by J.Isaacs and colleagues.

15. hMET/hHGF prostates appear to downregulate AR signaling – even in cells/tissues without full carcinoma phenotypes (Fig 2). This is not discussed.

16. It is not clear from the studies presented whether WNT pathway activation is needed in the context of hMET/hHGF to promote DNPC, nor whether hMET/hHGF is required in the context of activating Wnt signaling for DNPC.

17. The mechanism(s) by which XP01 and Selinexor exert anti-cancer effects in the DNPC tumors that lack AR activity have not been demonstrated in this study (since the DNPC

cells/tumors lack AR – some other XPO1 mechanism is likely operative).

18. The description of the tumors of primary PCa and ADT-treated mCRPC tissues are not well described in text/results – number of patients, tumor phenotypes, tumor purity, metastatic sites, etc.

19. The key corollary human cell/tumor experiments are lacking. Key experiments to be performed would include: (1) expressing HGF/MET in ARPC cells – determine if DNPC is promoted; (2) activating WNT in ARPC cells – determine if DNPC is promoted; (3) expressing XPO1 in ARPC – determine if DNPC is promoted, etc.

Minor Points

1. Supplemental Table 2 – it is not clear how the p-value and adjusted p-values for these analyses are almost identical in view of the large number of genes evaluated. Q values should be reported.

2. Methods – Page 23 Line 519. The dissociation methods for the scRNAseq are not described in sufficient detail.

3. Page 1 Line 12 – it is not clear what the authors mean when they state “..new, unknown, and deadly disease.”?

Reviewer #2 (Remarks to the Author):

I have reviewed the manuscript and I find that the mouse models generated are useful to study AR and neuroendocrine (NE)-null prostate cancer. The conclusion of a causal role for HGF/MET and Wnt signaling in promoting hormone refractoriness and castrate resistant prostate cancer development seem valid. However, the model proposed in Fig 6i need further evidence. Specifically, XPO1-mediated nuclear export being upstream of ribosome biogenesis and also whether inhibition off XPO-1 can reduce localisation of AR and ribosomal proteins to the cytoplasm.

I suggest the following to substantiate the model:

The authors suggest that stabilization of β -catenin augments SP1-regulated XPO1 expression. This need to be validates with experiments such as knocking down SP1 and examining the effects on XPO-1 expression and AR levels in the nucleus.

The authors suggest that the induction of XPO1 can be the cause of lacking nuclear AR in Solid-PCa cells and this implicate a regulatory role of XPO1 in DNPC development. The authors should examine AR levels and its nuclear/ cytoplasmic localisation following selinexor treatment in organoid cultures.

The model in Fig 6I depicts XPO1 expression and activation to be upstream of “ribosome biogenesis”.

Ribosome biogenesis occurs in the nucleoli. Ribosomal proteins are produced in the cytoplasm and imported to the nucleoli to assemble with rRNAs. Pre-ribosomal subunits are then exported to the cytoplasm to form functional ribosomes. Inhibition of nuclear export inhibits ribosome maturation not “biogenesis”. The inductions of ribosomal proteins levels in the cytoplasm can be driven by MYC, which is a master regulator of ribosome biogenesis and was shown to be upregulated in Solid-PCa cells compared to Adeno-PCa cells (Fig. 4k). The authors show assess whether XPO1 inhibition leads to a reduction in global translation and/ or a decrease in nucleolar size.

Also, the quantitation of AgNOR staining in Fig3g should represents nucleolar size. The representative images don't show increased number of nucleoli as suggested in the graph.

Fig 2C,K and I require wild type samples are controls.

Specifically, I don't understand why the effects of cell cycle on the data are not removed. These usually are referred to as “regressing out the cell cycle”. In this paper, there is more clustering by cell cycle stage, rather than less. They defined cells in S phase as a different cell type to those in other phases. For example, in Extended Data Figure 2a,b, they have phenotyped G2M cells as basal epithelial and leukocytes, and G1 as luminal epithelial cells. The don't provide an explanation for this.

Reviewer #3 (Remarks to the Author):

This study demonstrated ADT induces aberrant activation of MET and Wnt/ β -catenin signaling pathways using public PCa datasets, developed a new model for prostate cancer, generated single-cell RNA-seq data, and identified XPO1 as a potential therapeutic target for CRPC. The study is well written and presented. The analyses are logical. Addressing the

following concerns may help strengthen this study. Some of the conclusions are overstated, especially “DNPC” phenotype. Experimental validation using human PCa models is lacking. The regulatory mechanisms for AR nuclear export need further exploration.

1. It would be much better to perform IHC staining by using serial sections in Fig. 1e. Importantly, some panels in naïve PCa group even did not exhibit classical naïve PCa pathology.

2. The author demonstrated aberrant activation of HGF/MET and in DNPC or AR-low cancer cells by analyzing clinical RNA profiling datasets. However, it seems that Ar expression positively correlated with hMETtg expression in Fig. 2f. Can the author explain this inconsistent phenotype?

3. The author should be very careful to draw such conclusion that “These lines of evidence explore a regulatory mechanism for HGF/MET signaling activating canonical Wnt pathways through PCa development”.

Quantification analyses for Fig. 2l are really required to demonstrate an increase in cytoplasmic and nuclear β -catenin expression in PCa cells with positive pMET staining, because it is hard for me to see such increase in the pictures. Personally, I don't think whether HGF/MET could activate Wnt or not will largely influence this study.

4. The author constructed a GEMM with aberrant HGF/MET and Wnt activation and demonstrated a “DNPC” phenotype. However, it seems that the “DNPC” phenotype here was not consistent with clinical DNPC phenotype. I suggest that DNPC should not be overstated in this study. I have some concerns that: (1) Are there any evidences supporting that human DNPCs also exhibit increasing nucleolar size and number? (2) Clinically, it is not AR nuclear export, but lack of AR expression is commonly observed in DNPCs. The author should carefully examine the AR expression at both protein level and RNA level. (3) More clinical evidences are required to support the “DNPC” phenotype in this study.

5. Ar expression should be included in Fig. 4b and Fig. 4d.

6. Targeting the AR signaling axis has been the mainstay of PCa therapy over decades. It is surprising to see a PCa GEMM with obvious AR nuclear export. It would largely strengthen this study to explore the mechanisms for AR nuclear export in the GEMM. The author proposed the AR nuclear export was induced by XPO1 activation. However, robust evidences were largely lacking. The author could perform genetical manipulation or pharmacological inhibition of XPO1 to explore the regulatory mechanisms for AR nuclear export.

7. This study is mainly limited by the lack of experimental validation using human prostate cancer cell lines. I understand the author established important mouse models. At least, the author should (1) Validate the role of XPO1 in DNPC cell lines, such as PC3 and DU145, both in vitro and in vivo; (2) Use human PCa datasets to explore whether human DNPCs also highly expressed XPO1.

Reviewer #4 (Remarks to the Author):

In this manuscript, the authors have addressed important questions that has been inadequately represented in the field during the last couple of years. Although almost all primary prostate cancer cells express androgen receptor and are dependent on androgens, androgen deprivation therapy fails in most patients who develop castration-resistant prostate cancer.

To adress this question, they carried out a series of careful and well thought out experiments involving a crop of mouse models.

The current study indicates that XPO1 inhibition, may be by selinexor, might be a therapeutic option for treating Double-Null Prostate Cancer.

This is an important result for clinicians and researchers in the field and those working outside the field.

Below are some specific minor suggestions that could improve the manuscript.

1. Page 3, line 50-52. Although the authors claim that Selinexor is FDA-approved they should cite the relevant literature.

2. The authors demonstrate differences between Adeno-PCa and Solid-PCa cells. The Adeno-PCa cells as depicted are morphologically smaller. This fact should be addressed.
3. Page 6, paragraph 2. The authors elegantly demonstrate that the numbers of nuclear pore complexes differ. However no thoughts were give about the reasons and if possibly the composition of the nuclear pore complexes might be affected and potentially differ in the cell types.
4. Figure 6. The annotation in the figure is errornous.
5. The authors should comment on the nuclear import of AR and the potential role of nuclear import in their model.
6. The authors should discuss what is know about XPO AR interaction.

The methodology is sound and meet the recent standards

In summary, I think the paper is acceptable after carefully addressing the minor issues given above.

REVIEWER COMMENTS

Reviewer #1 (Remarks to the Author):

Title: Androgen Deprivation Induces Double-Null Prostate Cancer via Aberrant Activation of Nuclear Exporting and Ribosomal Biogenesis by HGF and WNT Axes

Authors: Blinded

Summary: This manuscript reports a series of studies designed to identify and target molecular drivers of prostate cancers that lack both AR signaling and activity of the neuroendocrine (NE) pathway – termed double null prostate cancer (DNPC). Through mining/analyses of human prostate cancer datasets, the authors identified HGF/MET and WNT signaling as differentially increased in DNPC relative to ARPC. Genetically engineered mouse models (GEMMs) were developed that promote MET pathway signaling vis cMET and HGF expression, and Wnt pathway activation via apc loss. Mice with MET/HGF activation develop low penetrance prostate cancer and reduced Ar activity, whereas the addition of apc loss produces high penetrance prostate cancers with variable phenotypes and metastases. Key genes upregulated in these models include XPO1 and ribosome biogenesis. Inhibitors of these features reduced prostate cancer growth.

We greatly appreciate the Reviewer's time and effort to review this manuscript. In recent years, more diverse CRPC phenotypes have been observed in patients treated with the new generation of AR antagonists and inhibitors for blocking androgen synthesis. Specifically, a subpopulation of DNPC has been reported recently in patients treated with abiraterone (ABI) and enzalutamide (ENZ). In this study, through analyzing relevant clinical datasets/samples, we identified aberrant activation of HGF/MET pathways and increased Wnt/ β -catenin pathways in human DNPC cells. To directly assess aberrant co-activation of HGF/MET and Wnt/ β -catenin pathways in prostate tumorigenesis, we devoted significant resources and effort to develop a series of new and relevant GEMMs, including *hHGFtg:H11^{hMET/+}:PB^{Cre4}* (DoubleTg) and *hHGFtg:H11^{hMET/+}:Ctnnb1^{L(Ex3)/+}:PB^{Cre4}* (TripleTg). These GEMMs comprise the center of this investigation. However, we unfortunately found that the Reviewer mistakenly stated the “*apc loss*” used in our mouse models. In fact, we used the stabilized β -catenin, a “gain-of-function” model, in this study. We sincerely hope this factual error will not create any misunderstanding and negatively affect the review process.

Comments:

1. Overall, the manuscript is well written and clear. The methods are generally described with sufficient detail that should allow reproducibility. The references cited are appropriate, though there are key omissions (see points 3 and 4).

Again, we greatly appreciate the Reviewer's positive comments on this study. We have carefully updated appropriate references as the Reviewer referred to below (see Points 3 and 4).

2. The novel aspects of the present study center on the development of several GEMMs that induce primary and metastatic prostate cancer. These studies indicate that HGF/MET and WNT signaling promote aspects of prostate cancer. These pathways have previously been shown to promote adverse prostate cancer behavior or directly cause prostate cancer in preclinical models and the present work confirms and further extends these observations with detailed single cell based analyses and comparative gene expression studies that nominate potentially druggable pathways.

Again, we thank the Reviewer for the positive comments regarding the clinical relevance of this study. We agree with the Reviewer that our data of identifying aberrant co-activation of HGF/MET and WNT/ β -catenin pathways in prostate cancer progression and DNPC development

provides mechanistic insight into our current knowledge on ADT-induced CRPC development, and may lead to the development of new inhibitors and therapeutic strategies to efficiently target these oncogenic pathways for treating advanced prostate cancer in the near future.

3. With respect to cMET - prior studies have reported that AR represses MET expression and AR knockdown or tumors with low AR expression activate/upregulate MET expression and activity. (PMID: 24478054; PMID: 23877345; PMID: 20946682; PMID: 26806347; others). These prior reports reduce the novelty of the present work. These studies do not appear to have been referenced.

We appreciate the Reviewer's comment, and were well aware of those prior studies. However, DNPC is a relatively new disease and, in fact, the molecular mechanisms underlying its pathogenesis are still largely unknown. To directly address the reviewer's comment, we updated and added the related references in the current revision (Page 18, lines 400-403) and provided more rationale for the significance of the current study that is solely focused on DNPC pathogenesis (Pages 2, lines 39-41; lines 41-44; Page 5, lines 90-92).

4. With respect to WNT signaling – prior studies have induced DNPC and PC metastases via Apc loss in mouse models, and demonstrated that WNT pathway antagonists repress PC growth - also reducing the novelty of the present findings. These studies do not appear to have been referenced (example: (PMID: 32217460; PMID 32376773).

Again, the “Apc loss” model was not used in this study, as mentioned above. We are also aware of the referred reports. Specifically, we are very familiar with and appreciate the previous study (PMID: 32376773) using *p53*, *Pten*, and *apc* deficient compound mice. Although *apc* deficiency and β -catenin GOF both increase Wnt signaling, there were many biological differences between these GEMMs. For example, loss of APC has been shown to have additional effects on the mitotic spindle and chromosomal instability. In response to the Reviewer's point, we revised the related text and re-emphasized the “stabilized β -catenin” mouse model used in this study (Page 7, lines 140-44).

5. The human data referenced/analyzed by the authors that associate HGF/MET and WNT with DNPC demonstrate that the findings are not unique to DNPC – these pathways are also active/differentially increased in NEPC as observed in Fig1a – this is not discussed.

We appreciate the Reviewer's point. In the above analyses, we identified a similar enrichment in the down-regulation of androgen-response pathways in DNPC and NEPC in comparison to ARPC samples. However, no significantly enriched HGF/MET and Wnt/ β -catenin signaling pathways were identified between NEPC and ARPC. These data directly address the Reviewer's comments. We provided the relevant data and modified the text in this revision (see the modified Supplementary Fig.1b, and pages 3-4, lines 62-67).

6. The transgenic models with hMET + hHGF generate primarily PIN lesions – not carcinomas. Overt carcinoma appears to be quite rare. It is not clear what ‘intracystic’ prostate cancer is? This terminology is not used in assessing primary human prostate cancer. The histopathology should be submitted to expert human pathologists for analysis. Detailed images of these tumors should be provided. Even with aging – 9-12 months only 2/11 mice had some evidence of carcinoma.

We fully appreciate that murine neoplasms in many organs including in the prostate are pathologically, biologically, and clinically different from their human counterparts. Therefore, we have been extremely careful on pathological analyses with the mouse models reported in this study. All of the pathological analyses in this study were carefully evaluated and reported by two experienced and board-certified human GU surgical pathologists independently, who also have excellent track-record in mouse prostate cancer models. Specifically, in this study, we carefully identified both mouse PIN and “intracystic” tumor lesions in the hMET/hHGF transgenic mice (DoubleTg mice) based on

the guidelines provided by “the New York Pathology Panel” (see PMID: 23610450). The term “intracystic carcinoma” of prostate in mouse models was suggested at “The Mouse Models of Human Cancers Consortium Prostate Pathology Committee” in 2013. The mouse “intracystic” prostate tumors are characterized by proliferating masses of atypical cells with marked expansion of preexisting structures without obvious infiltrative growth (see Figure 3 of PMID: 23610450). These lesions were only observed in mice but not in humans. We also provided more pathological images of the transgenic mice in this revision in response to the Review’s other comments (Please see the revised Supplementary Fig. 2b).

7. The Wnt pathway activation by hMET+hHGF appears to be insufficient to robustly induce carcinoma. Genetic manipulation to activate Wnt signaling – did result in more substantial carcinomas and notably metastases were induced. These tumors exhibited a mixed phenotype termed by the authors as adenocarcinoma and ‘solid’ prostate cancer. ‘Solid’ prostate cancer is not a generally accepted histopathological term. The authors should consult with a human prostate pathology expert to establish appropriate nomenclature and determine relevance to human PC.

Again, in this study, two board-certified human GU surgical pathologists who also have an excellent track-record on mouse prostate cancer models worked closely with us to analyze the mouse tumor samples. A series of progressive prostate tumor lesions were observed during age-progression in TripleTg mice. They included the lesions with glandular characteristics and well-differentiated adenocarcinomas, similar to human Gleason Grade 3-4 prostate carcinomas (termed Adeno-PCa) and the lesions with poorly-differentiated characteristics containing abundant lightly eosinophilic cytoplasm and pleomorphic nuclei without distinct gland formation, akin to Gleason Grade 5 prostate carcinomas, termed Solid-PCa). Based on the Bar Harbor Classification of mouse prostate tumors (PMID: 15026373), the "Adeno-PCa" can be classified as well differentiated (3.2.1.1), and the "Solid-PCa" as poorly differentiated (3.2.1.3) adenocarcinomas. Importantly, we used these two abbreviated terms **only** for this study but not to make them as the general “histopathological terms”. We specifically addressed this point in this revision to avoid confusion (see Page 7-8, lines 160-61; Page 19, lines 422-23).

8. Page 8 Line 162 – the authors state that ‘lack of nuclear AR expression in poorly differentiated Solid-PCa cells corroborates our observations in human DNPC...’ – it is not clear where human DNPC samples were stained for AR in this present study? Showing the murine DNPC next to the human DNPC histology should be performed in a statistical manner to confirm the correlations.

In fact, we presented those referred human data in the bottom panel of the original Figure 1e, in which the lack of typical nuclear AR staining is revealed in ABI- and ENZ-treated CRPC patient samples (Page 4, lines 71-77). In this revision, we also measured the numbers of positive staining of AR and SYN in different fields of three different slides in both human and TripleTg mouse samples (see Supplementary Fig. 3d and Page 8, lines 176-79) to directly address the Reviewer’s comment.

9. The prostates of ‘TripleTg’ mice are shown to exhibit 2 different phenotypes – the ‘adeno’ and the ‘solid’ – but presumably both types and the cells of these types harbor the same constellation of transgene events – it would be useful to understand what is driving the diversity in phenotype and gene expression.

We appreciate this insightful point. As shown in the “TripleTg” mice, co-activation of HGF/MET and WNT/ β -catenin pathways produced the faster growing and more aggressive tumor phenotypes. Pathological examination identified a series of progressive tumor lesions in different aged mice, including intracystic adenocarcinoma, invasive adenocarcinoma, and invasive carcinoma. Specifically, prostate tumor lesions with either glandular characteristics termed Adeno-PCa, or poorly differentiated cell element and lacking distinct gland formation, termed Solid-PCa, were identified at

10-month-old TripleTg mice. The observations of “Adeno-PCa” prior to the “Solid-PCa” development further demonstrated the progressive nature of these tumor lesions, implicating a regulatory mechanism underlying co-activating HGF/MET and WNT/ β -catenin pathways in promoting tumor progression. Therefore, in this study, we performed a series of experiments to identify the “driving” force regulated by HGF/MET and WNT/ β -catenin pathways to advance tumor progression. First, significantly increased nucleolar size and number, as well as more nuclear pore complexes (NPC) were identified in “Solid-PCa” cells in comparison to “Adeno-PCa” cells (Fig. 3g-j), directly demonstrating their fast growing and rapid cell-cycle progression features. Data from scRNA-seq analyses further showed higher expression of cell proliferation genes in the “Solid-PCa” than the “Adeno-PCa” cells, providing the molecular basis for their fast-growing and poorly differentiated tumor cell properties (Fig. 4f). Moreover, single-cell trajectory analyses uncovered a series of significantly enriched signaling pathways that directly promote tumor growth and progression in the “Solid-PCa” cell clusters. Including protein synthesis, translation, rRNA processing, nuclear exporting, ribosome biogenesis, and epithelial-mesenchymal transition (EMT) activation, etc (Fig. 4j). In response to the Reviewer’s comments, we further modified the related figures and text to specifically emphasize the mechanistic significance of our findings: identifying the critical role of co-activating HGF/MET and WNT/ β -catenin pathways in driving tumor progression and DNPC development (see the revised Fig. 7i).

10. Page 11 Line 233 – the authors state that XPO1 and RPL12 are detected in human DNPC samples from ABI- and ENZ treated patients. While notable, (i) the conclusions and validation are not statistically based; and (ii) AR and AR target genes NE genes are not shown. Further – lacking from these samples are data demonstrating HGF/MET or WNT activity in these DNPC tumors – in comparison to ARPC and NEPC. Showing such associations would greatly strengthen the conclusions as clinically relevant.

In this study, using IHC approaches, we identified increased XPO1 and RPL12 in human samples from ABI and ENZ treated patients (Supplementary Fig. 5a). Additionally, the peri-nuclear staining of XPO1 appeared only in CRPC samples from ABI- and ENZ-treated patients, which is very similar as observed in Solid-PCa cells of TripleTg mice (Fig. 4k). These clinical samples also showed lacking AR and SYN expression with increased pMET and nuclear/cytoplasmic β -catenin (see Fig. 1e). In this revision, we quantified AR and SYN expression in prostate tumor cells of clinical samples and TripleTg mice, and identified significantly increased AR-SYN- cells in human CRPC cells and mouse Solid-PCa cells in comparison to naïve PCa and Adeno-PCa cells, respectively (Supplementary Fig. 3d). Moreover, our new GSEA further showed significantly increased HGF/MET or WNT activity only in DNPC but not in NEPC in comparison to ARPC (Fig. 1b and Supplementary Fig. 1b, and Pages 3-4, lines 62-67). These data directly address the Reviewer’s concerns, and have been incorporated into this revision.

11. Similarly – the analyzed human RNAseq data showing XPO1 expression in two mCRPC epithelial clusters lacks any description/association with AR activity/DNPC phenotypes or in the context of WNT or HGF/MET activity (Expend Data Fig 5b).

As shown in Extend Data Fig. 5b, we demonstrated increased XPO1 expression in two mCRPC cell clusters. Actually, we also analyzed the same set of clinical samples for the AR and NE scores and the expression/activation of AR, MET and Wnt downstream targets, which were placed in Supplementary Fig. 1h-i). Analyses of both AR and NE scores in these cell clusters showed their DNPC cellular properties. In the current revision, we modified the related text and figures to explicitly highlight this data based on the Reviewer’s comments (Pages 4-5, lines 84-88; Page 13, lines 275-77)

12. Page 15 line 333 – the authors suggest that XPO1 is a new and potential target for treating DNPC – first, XPO1 is not a new target for prostate cancer overall, and has been evaluated in the context of co-targeting AR in clinical studies where: a) it had very limited effects; and b) there was poor tolerance (PMID: 29487219). Targeting cMet has also had very limited efficacy (PMID: 32943461). This reduces the novelty and potential clinical impact of the findings.

We appreciate the Reviewer's comment. We are also aware of the previous study with Selinexor in mCRPC patients and indeed cited it in the original manuscript. Actually, the previous study demonstrated clinical activity but also poor tolerability in mCRPC patients, limiting the future clinical application (PMID: 29487219). Additionally, in the same vein, the previous clinical study cited by the Reviewer also showed that concurrent administration of both Enz and Met inhibitor, crizotinib, resulted in a significant decrease in systemic crizotinib exposure in patients (PMID: 32943461). These studies actually do not reduce the significance of the scientific concepts for those therapeutic strategies, but, in contrast, support further research into the development of more selective, potent and synergistic inhibitors in these areas. Thus, our data presented in this current study shed fresh light on the development of new inhibitors and therapeutic strategies that not only can specifically target these oncogenic pathways but also possess less side effects for patients. In the current revision, we have provided more discussion to specifically address these points of the Reviewer (Page 20, lines 442-45; Pages 20-21, lines 454-61).

13. The mechanism of Selinexor and XPO1 in the context of the model proposed is not clear. If AR loss is mediated by XPO1 (nuclear export)– then does XPO1 inhibition restore nuclear AR? If AR is lost by some other mechanism – then what is Selinexor inhibition of XPO1 doing? Clearly, other transcription factors may be inhibited by XPO1 blockade – this is not specific to AR.

We greatly appreciate this insightful and important point. Dysregulation of XPO1 has been reported in many different human malignancies. Emerging evidence also showed that XPO1 can regulate more than 200 proteins and molecules to promote tumor growth and progression. In this study, we identified that aberrant co-activation of HGF/MET and WNT/ β -catenin induces XPO1 expression/activity in Solid-PCa cells of TripleTg mice. Using prostatic organoid culture systems, we observed that XPO1 inhibition by Selinexor showed a significant repressive effect on PCa cell growth in the samples cultured either with or without DHT. Given nuclear AR expression was only revealed in PCa cells cultured with DHT, these data suggest the inhibitory effect of XPO1 on PCa cells is not only for AR-mediated oncogenic pathways. Additionally, Selinexor has shown inhibitory effects on other human malignancies that are not regulated by AR signaling pathways. Moreover, AR nuclear expression has been shown to be regulated by the nuclear import signaling pathways as well (see Reviewer 4 points below). Therefore, we agree that XPO1 blockade can repress other transcription factors and is not specific to AR in PCa cells. In this revision, we also showed the effect of XPO1 inhibition on reducing both global translation and decreasing nucleolar size and number in PCa cells derived from TripleTg mice (Supplementary Fig. 7a-d). These data further suggest the important role of XPO1 in facilitating rRNAs, RPs, and assembly factors in ribosomal biogenesis and protein synthesis during tumor growth and progression. In this revision, we made corresponding changes in response to the Reviewer's comment (Page 20, lines 439-45). Therefore, this current study will lead to more investigations using our current GEMMs to further address these important questions in the field.

14. Page 15 Line 338 – the authors state that there are no biologically relevant in vivo models to investigate DNPC pathogenesis. This is not correct. See PMID 32376773. There are also PDX lines that are DNPC reported by J. Isaacs and colleagues.

Unfortunately, we were unable to find the PDX lines in the report authored by Issacs as referred by the Reviewer (PMID 32376773). The Reviewer might overlook this. Nonetheless, we have modified the above sentence in the current revision (Page 18, lines 392-94).

15. hMET/hHGF prostates appear to downregulate AR signaling – even in cells/tissues without full carcinoma phenotypes (Fig 2). This is not discussed.

We apologize for the confusion, but, actually, there is no data to show that hMET/hHGF down-regulates AR signaling in the prostate of our GEMMs in the previous Fig. 2. As described in the manuscript, it has shown that the AR can repress the transcription of the *Met* gene through interfering with SP1 binding to the endogenous promoter of the *Met* gene (PMID: 17283128). In response to the Reviewer's comment, we carefully examined this revision to make sure those points are more explicitly made in the manuscript. Specifically, we provided the AR expression plots in the revised Fig.4b in the current revision.

16. It is not clear from the studies presented whether WNT pathway activation is needed in the context of hMET/hHGF to promote DNPC, nor whether hMET/hHGF is required in the context of activating Wnt signaling for DNPC.

One of the most important findings in this study is to identify DNPC development only in the TripleTg mice, *hHGFtg:H11^{hMET/+}:Ctnnb1^{L(Ex3)/+}:PB^{Cre4}*, but neither the DoubleTg, *hHGFtg:H11^{hMET/+}:PB^{Cre4}*, nor stabilized β -catenin only mice, *Ctnnb1^{L(Ex3)}:PB^{Cre4}*. These data directly demonstrated the critical role of co-activating HGF/MET and Wnt/ β -catenin signaling pathways to promote DNPC tumorigenesis. Accordingly, in this study, we also provided multiple lines of experimental evidence to demonstrate the aberrant activation of HGF/Met signaling pathways to induce Wnt/ β -catenin activation, providing the scientific rationale and biological relevance for investigating the co-activation of HGF/MET and Wnt/ β -catenin in prostate cancer progression and DNPC development. While we fully appreciate the Reviewer's insight points, our data demonstrated the collaborative role by HGF/MET and Wnt/ β -catenin co-activation in promoting DNPC development. In this revision, we made the above data explicitly clear in addressing the Reviewer's point. We also modified the related text in the "Discussion" to further clarify our points (Page 18, lines 396-400; Page 19, lines 425-431).

17. The mechanism(s) by which XPO1 and Selinexor exert anti-cancer effects in the DNPC tumors that lack AR activity have not been demonstrated in this study (since the DNPC cells/tumors lack AR – some other XPO1 mechanism is likely operative).

We agree with the Reviewer. As we explained above (see Point #13), the critical role of XPO1 in promoting tumor progression has been demonstrated in a variety of human malignancies that are not regulated through AR signaling pathways. It has also been shown that XPO1 regulates the nuclear export of more than 200 proteins and molecules in tumor cells. These lines of evidence clearly suggested that a critical role of XPO1 in tumor progression and drug resistance is mediated through various oncogenic pathways rather than a single route. The lack of AR expression/activity has been observed in the "Solid-PCa" cells of the "TripleTg mice", and thus the inhibitory effect of Selinexor on these tumor cells is unlikely to be directly regulated through the AR. Again, we have carefully modified the current manuscript to make sure the above points are clear and adequately represented. Additionally, more efforts are clearly needed to be devoted to better and completely understand the regulatory role of XPO1 in prostate tumorigenesis and CRPC development. These points have been re-emphasized in the current revision (Pages 20-21, lines 454-61, Pages 21, lines 471-73).

18. The description of the tumors of primary PCa and ADT-treated mCRPC tissues are not well described in text/results – number of patients, tumor phenotypes, tumor purity, metastatic sites, etc.

In response to the Reviewer's comment, we have provided more and detailed information regarding the above clinical samples that were used in this study (see Page 23, lines 509-14). Briefly, we used four prostatectomy samples from 3 different patients without hormonal treatment and metastatic castration-resistant prostatic carcinoma specimens from 6 patients who had received second line antiandrogen therapies- ENZ and/ or ABI in this study. The metastatic tumor samples were of high tumor purity.

19. The key corollary human cell/tumor experiments are lacking. Key experiments to be performed would include: (1) expressing HGF/MET in ARPC cells – determine if DNPC is promoted; (2) activating WNT in ARPC cells – determine if DNPC is promoted; (3) expressing XPO1 in ARPC – determine if DNPC is promoted, etc.

In recent years, more diverse CRPC phenotypes have been observed in patients who received current ADT. In this study, we directly addressed this extremely significant challenge, and investigated the regulatory mechanisms underlying current ADT induced tumor progression and DNPC development. Our data from analyzing relevant clinical samples showed that current ADT enhances HGF/MET expression/activity, which further induces Wnt/ β -catenin activation to promote tumor progression. Using a series of newly generated GEMMs, we further showed that aberrant co-activation of HGF/MET and Wnt/ β -catenin further elevates XPO1 and ribosomal biogenesis signaling pathways to advance tumor progression, metastasis, and DNPC development. These data demonstrate the collaborative, reconcilable, and consequential effects of these different oncogenic pathways on promoting and advancing DNPC development during the course of tumor progression initiated by current ADT, rather than the result of a separate, single, and independent alteration and dysregulation through the individual pathway(s). Given the above lines of scientific evidence, we respectfully believe that future efforts should focus on the collaborative role of those oncogenic pathways in DNPC pathogenesis using biologically relevant and *in vivo* model systems. In this study, we have also given our best effort to evaluate and validate the clinical relevance of XPO1 and ribosomal biogenesis activation in PCa progression and DNPC development using current available patient samples and clinical datasets. In this revision, we specifically emphasized these collaborative pathway interactions induced by current ADT to promote tumor progression and metastasis, and DNPC development in the "Introduction", "Results" and "Discussion" sections of the revision.

Minor Points

1. Supplemental Table 2 – it is not clear how the p-value and adjusted p-values for these analyses are almost identical in view of the large number of genes evaluated. Q values should be reported.

In response to the Reviewer's comment, we carefully examined the both p-value and adjust p-values on the Supplemental Table 2. They do not appear identical and are appropriately represented. In response to the Reviewer's point, we added "Q-value" to the revised Supplementary Data_2-6.

2. Methods – Page 23 Line 519. The dissociation methods for the scRNAseq are not described in sufficient detail.

We have provided more and detailed information for the dissociation methods in response to the Reviewer's comment above (see Page 28, lines 614-25)

3. Page 1 Line 12 – it is not clear what the authors mean when they state "new, unknown, and deadly disease."?

We modified this sentence in the current revision to respond to the Reviewer (see Page 1, lines 10-11).

Reviewer #2 (Remarks to the Author):

I have reviewed the manuscript and I find that the mouse models generated are useful to study AR and neuroendocrine (NE)-null prostate cancer. The conclusion of a causal role for HGF/MET and Wnt signaling in promoting hormone refractoriness and castrate resistant prostate cancer development seem valid. However, the model proposed in Fig 6i need further evidence. Specifically, XPO1-mediated nuclear export being upstream of ribosome biogenesis and also whether inhibition off XPO-1 can reduce localisation of AR and ribosomal proteins to the cytoplasm. I suggest the following to substantiate the model:

We greatly appreciate the Reviewer's positive comments on our mouse models and works. Using a variety of new and relevant GEMMS and other *in vivo* models, multiple human datasets/samples, and different experimental and analytic approaches, we identified a regulatory mechanism by which current ADT induces various oncogenic pathways and regulators to promote PCa progression and DNPC development. To make our data clearer to the Reviewers and readers, we presented a working model (Fig. 6i, the current Fig. 7i) to summarize our major findings and imply the potential regulatory mechanisms open for the future further investigations. Given the nature of such hypothetical models, we also explore several important signaling pathways and regulators for ADT induced disease progression and DNPC development. In response to the Reviewer's comment, we have modified the figure and related text in the current revision. Specifically, we made changes to XPO1-mediated nuclear export being upstream of ribosome biogenesis and the effect of XPO-1 on AR cellular localization in the current Fig. 7i. We feel the revised model adequately summarizes our current data and suggests the directions for the future experiments based on the current literature.

1) The authors suggest that stabilization of β -catenin augments SP1-regulated XPO1 expression. This need to be validates with experiments such as knocking down SP1 and examining the effects on XPO-1 expression and AR levels in the nucleus.

We appreciate the Reviewer's point. The expression of Sp1 has been shown in various prostate cancer cell lines (PMID: 29427323). A regulatory role of Sp1 on the transcription of Xpo1/Crm1 has also been identified. Specifically, multiple Sp1 binding sites were identified within the promoter region of the *Xpo1/Crm1* gene, and, through these sites, Sp1 can regulate *Xpo1/Crm1* transcription in transformed tumor cells (PMID: 21683812). In this study, we further demonstrated the regulatory role of Sp1 on Xpo1 transcription in prostate tumor cells of the TripleTg mice using ChIP-qPCR analyses, which is consistent with the previous reports and implicating a new mechanism underlying aberrant XPO1 expression in DNPC cells. To directly address the Reviewer's comments, we performed the knockdown experiments using human prostate cancer cell lines and prostatic tumor organoid cells to examine the effect of Sp1 on XPO1 expression. As shown in the revised Fig. 5d and 5e, a significant reduction of XPO1 mRNA transcript and protein expression was observed in Sp1 knockdown samples in comparison to the controls.

2) The authors suggest that the induction of XPO1 can be the cause of lacking nuclear AR in Solid-PCa cells and this implicate a regulatory role of XPO1 in DNPC development. The authors should examine AR levels and its nuclear/cytoplasmic localisation following selinexor treatment in organoid cultures.

Again, we greatly appreciate the Reviewer's point. In this study, we demonstrated aberrant activation of XPO1 and ribosome biogenesis in Solid-PCa cells of TripleTg mice and human DNPC cells. However, as explained above, we fully appreciate the fact that XPO1 regulates the nuclear export of many different proteins and molecules in a variety of human malignancies, many of which are actually not involved in AR signaling pathways. Additionally, as indicated in the manuscript, the AR is a nuclear hormone receptor and its nuclear localization is primarily regulated by androgens. Therefore, current ADT blocks AR nuclear localization to inhibit AR-regulated oncogenic growth. In

response to the Reviewer's comment, we examined the cellular localization of AR in Selinexor treated samples that were cultured either with DHT or without DHT. The significant inhibition of Selinexor on cell growth was observed in the both samples. However, most tumor organoids cultured in the absence of DHT showed no AR expression, and there is no significant difference in AR expression between Vehicle and Selinexor treated samples cultured in the absence of DHT. Additionally, there was also no significant difference in AR nuclear expression between the samples treated with Vehicle and Selinexor and cultured with DHT. Using co-IF approaches, we also observed more intense peri-nuclear staining of XPO1 corresponding to less AR nuclear staining in Solid-PCa than Adeno-PCa cells of TripleTg samples (Supplementary Fig. 7e). These results further suggest that both nuclear import and export mechanisms contribute to AR nuclear expression. In this revision, we also provided more discussion on those points (Pages 22-23, lines 498-501).

3) The model in Fig 6i depicts XPO1 expression and activation to be upstream of "ribosome biogenesis". Ribosome biogenesis occurs in the nucleoli. Ribosomal proteins are produced in the cytoplasm and imported to the nucleoli to assemble with rRNAs. Pre-ribosomal subunits are then exported to the cytoplasm to form functional ribosomes. Inhibition of nuclear export inhibits ribosome maturation not "biogenesis". The inductions of ribosomal proteins levels in the cytoplasm can be driven by MYC, which is a master regulator of ribosome biogenesis and was shown to be upregulated in Solid-PCa cells compared to Adeno-PCa cells (Fig. 4k). The authors show assess whether XPO1 inhibition leads to a reduction in global translation and/ or a decrease in nucleolar size. Also, the quantitation of AgNOR staining in Fig3g should represents nucleolar size. The representative images don't show increased number of nucleoli as suggested in the graph.

We again thank the Reviewer for these insightful comments. To directly respond to these points, we performed a series of additional experiments/analyses during the past months, including 1) to examine the effect of XPO1 inhibition on Global translation in human prostate cancer cell lines, 2) to measure the nucleolar area in both mouse Solid tumor cells and human DNPC cells, and 3) to analyze the numbers of individual AgNOR dots for nucleoli in the above mouse and human tumor samples, and 4) to assess the effect of Selinexor on nucleolar size in PCa cells implanted *in vivo*. Data from the above new experiments/analyses have been included in the current revision to directly respond to the Reviewer comments (see the revised Supplementary Fig. 7a,b, Supplementary Fig. 3g-i, Fig. 3g, and Supplementary Fig. 7c,d). They are consistent with our previous results and provided additional experimental evidence demonstrating nucleolar abnormalities in both mouse and human PCa lesions. Additionally, we also specifically addressed the regulatory role of MYC in aberrant ribosomal activation in DNPC-like tumor cells (see Pages 12-13, lines 269-73). Based on these new data, we also updated the hypothetical model presented in the current Fig. 7i and modified the related text in response to the Reviewer's comment.

4) Fig 2C, K and I require wild type samples are controls.

In response to the Reviewer's comment, we provided all control samples from wild type mice for Fig 2c, k, and l. Please see the current Supplementary Fig. 2a, Fig. 2k, and Supplementary Fig. 2m, n, respectively.

5) Specifically, I don't understand why the effects of cell cycle on the data are not removed. These usually are referred to as "regressing out the cell cycle". In this paper, there is more clustering by cell cycle stage, rather than less. They defined cells in S phase as a different cell type to those in other phases. For example, in Extended Data Figure 2a,b, they have phenotyped G2M cells as basal epithelial and leukocytes, and G1 as luminal epithelial cells. The don't provide an explanation for this.

We apologize for the confusion. In the scRNA-seq analyses in this study, we first performed the "cell cycle phase regression" to remove the effects of cell cycle genes before making cell clusters

and performing other analyses. Due to the size and color of dots that were used for representing different cell cycles in the original plots, certain cell clusters appeared to be separated by G2M status, while cells in these clusters actually demonstrated mixed cell cycle status. In response to the Reviewer's comment, we made the changes and provided the new plots with different colors and sizes of dots to depict the data more clearly and accurately (please see the modified Supplementary Fig. 2c, 2h).

Reviewer #3 (Remarks to the Author):

This study demonstrated ADT induces aberrant activation of MET and Wnt/ β -catenin signaling pathways using public PCa datasets, developed a new model for prostate cancer, generated single-cell RNA-seq data, and identified XPO1 as a potential therapeutic target for CRPC. The study is well written and presented. The analyses are logical. Addressing the following concerns may help strengthen this study. Some of the conclusions are overstated, especially "DNPC" phenotype. Experimental validation using human PCa models is lacking. The regulatory mechanisms for AR nuclear export need further exploration.

We truly appreciate the Reviewer's positive comments on our work. To address the Reviewer's other comments, we performed a series of additional experiments using human cells and relevant datasets, and also carefully revised the related figures and text in corresponding to additional data. Additionally, we also toned down the "DNPC" phenotypes in the current manuscript as the Reviewer suggested. We also addressed the reviewer's other points below.

1. It would be much better to perform IHC staining by using serial sections in Fig. 1e. Importantly, some panels in naïve PCa group even did not exhibit classical naïve PCa pathology-

To directly address the Reviewer's comment, we collected relevant patients' samples and repeated IHC analyses using serial sections. The current images also exhibit the pathology of human naïve prostate cancer (see the current Fig. 1e).

2. The author demonstrated aberrant activation of HGF/MET and in DNPC or AR-low cancer cells by analyzing clinical RNA profiling datasets. However, it seems that Ar expression positively correlated with hMETtg expression in Fig. 2f. Can the author explain this inconsistent phenotype?

We apologize for the confusion. As shown in Fig. 2a, a loxP-flanked transcriptional silencing element, *LoxP-stop-loxP* (*LSL*) cassette, was inserted between the CAG promoter and the human *MET* transgene, *hMETtg*. Because the CAG promoter is ubiquitously active in mouse tissues, the *hMETtg* expression can be achieved in a constitutive but tissue-specific manner through *Cre* recombinase-mediated removal of the *LSL* cassette. Using the modified probasin promoter, an AR/androgens-regulated promoter, driven *Cre*, the expression of the *hMETtg* can be activated in AR positive luminal cells. In this revision, we provided related information for the above characteristics in regulating the expression of *hMET* transgene in the transgenic mouse model (see Page 24, lines 536-39).

3. The author should be very careful to draw such conclusion that "These lines of evidence explore a regulatory mechanism for HGF/MET signaling activating canonical Wnt pathways through PCa development". Quantification analyses for Fig. 2l are really required to demonstrate an increase in cytoplasmic and nuclear β -catenin expression in PCa cells with positive pMET staining, because it is hard for me to see such increase in the pictures. Personally, I don't think whether HGF/MET could activate Wnt or not will largely influence this study.

Again, we appreciate the Reviewer's comment. To directly address the Reviewer's concern, we quantified the numbers of both pMET and nuclear/cytoplasmic β -catenin positive cells in *hHGFtg:H11^{hMET/+}:PB^{Cre4}*, *DoubleTg* mice, and wild type control mice. As shown in the current

Supplementary Fig. 2n, increased pMET and cytoplasmic/nuclear β -catenin double positive cells revealed in PCa cells in comparison with the controls. Moreover, in this study, we also provided multiple lines of additional experimental evidence to demonstrate the regulatory role of HGF/MET activation in inducing canonical Wnt pathways during the course of prostate tumor development in our doubleTg mice. The scRNA-seq analyses showed increased Wnt/ β -catenin downstream target gene expression and significant enrichment of Wnt/ β -catenin signaling pathways in hMET+ cells (Fig. 2g and 2h). Moreover, co-expression of *hMETtg* and β -catenin target genes was significantly correlated in hMET+ cells (Fig. 2j). Quantitative reverse transcription-PCR (qRT-PCR) analyses further demonstrated higher expression of Wnt/ β -catenin targets, *Cd44*, *Sox9*, *Mmp7*, and *Plaur* in RNA samples prepared from prostate tumor tissues of DoubleTg mice than those from controls (Fig. 2k). These data, in combination with the above IHC and Co-IF data, demonstrated aberrant activation of HGF/MET signaling in inducing canonical Wnt pathways.

4. The author constructed a GEMM with aberrant HGF/MET and Wnt activation and demonstrated a “DNPC” phenotype. However, it seems that the “DNPC” phenotype here was not consistent with clinical DNPC phenotype. I suggest that DNPC should not be overstated in this study. I have some concerns that: (1) Are there any evidences supporting that human DNPCs also exhibit increasing nucleolar size and number? (2) Clinically, it is not AR nuclear export, but lack of AR expression is commonly observed in DNPCs. The author should carefully examine the AR expression at both protein level and RNA level. (3) More clinical evidences are required to support the “DNPC” phenotype in this study.

We appreciate the Reviewer’s comments regarding the biological relevance of the mouse model. We are also well aware that developing biologically relevant prostate cancer mouse models is always difficult and challenging. Specifically, the gross anatomy and histology of murine prostates are very different from those in humans. In fact, significant pathological, biological, and clinical differences have been reported in many murine neoplasms developed from different organs/tissues in various GEMMs in comparison to their human counterparts, including those in the prostate. In this study, we have been extremely careful in examining the phenotypes of our newly developed TripleTg mouse model and gave our best effort to appropriately describe the gross and pathological changes. In this revision, we directly addressed the Reviewer’s concerns by performing a series of additional experiments/analyses. First, we examined the nucleolar size and numbers in human prostate cancer samples from the patients treated with both ENZ and ABI (Supplementary Fig. 3h,i). Second, we assessed the RNA and protein levels of the AR in human CRPC samples and related datasets (the current Fig. 1c-e, Supplementary Fig. 1h-i, and Supplementary a Fig. 5c). Lastly, we validated the activation of HGF/Met and Wnt/ β -catenin signaling pathways and upregulation of XPO1 and ribosomal proteins in human DNPC samples using current human datasets (Supplementary Fig. 5c-d). Data generated from these new experiments/analyses are consistent with our findings and also directly address the Reviewer’s comments. They have been included in the current revision.

5. Ar expression should be included in Fig. 4b and Fig. 4d.

To address the Reviewer’s comment, we included Ar expression plots in the revised Fig. 4b, and 4d. We modified the related text corresponding to these changes. In the current revision, we have addressed the findings (see Page 10, lines 217-20; Page 11, lines 227-30).

6. Targeting the AR signaling axis has been the mainstay of PCa therapy over decades. It is surprising to see a PCa GEMM with obvious AR nuclear export. It would largely strengthen this study to explore the mechanisms for AR nuclear export in the GEMM. The author proposed the AR nuclear export was induced by XPO1 activation. However, robust evidences were largely lacking. The author could

perform genetical manipulation or pharmacological inhibition of XPO1 to explore the regulatory mechanisms for AR nuclear export.

We greatly appreciate this insightful comment. As described above, more diverse CRPC phenotypes have been developed in patients treated with more potent AR antagonists and inhibitors for blocking androgen synthesis recently. Specifically, a subpopulation of DNPC has been observed frequently in patients treated with ABI and ENZ. To directly address this new challenge in the field, we analyzed the relevant clinical datasets and samples, and identified aberrant activation of HGF/MET pathways and elevated Wnt/ β -catenin pathways in DNPC cells. Based on these data, we developed a series of relevant GEMMs to further assess aberrant co-activation of HGF/MET and WNT/ β -catenin pathways in prostate tumorigenesis. The TripleTg mice developed an early onset, fast growing, and very invasive tumor phenotype. Pathological analyses further showed poorly differentiated and the “DNPC” like cells in mouse prostate tumors. Our data from the both human and mouse samples and datasets demonstrate that aberrant activation of HGF/MET and Wnt/ β -catenin signaling pathways promotes tumor progression and induces DNPC development. Using scRNA-seq and other experimental approaches, we further identified altered Xpo1 and ribosomal biogenesis activation in DNPC-like cells in TripleTg mice, as well as in human samples. However, as explained above, we fully appreciate the complexity of the regulatory mechanisms underlying AR cellular localization by both nuclear import and export machineries. For example, current ADT directly abolishes AR nuclear localization through different regulatory pathways/regulators. Additionally, XPO1 regulates more than 200 proteins and molecules in various tumor cells to promote tumor growth and progression. Therefore, given DNPC development occurred after current ADT, it is unlikely that the lack of nuclear AR is solely regulated through XPO1-mediated pathways in DNPC cells. Moreover, our data also showed that XPO1 inhibition significantly repressed the growth of PCa cells with or without nuclear AR expression and cultured in the presence or absence of DHT, respectively. Furthermore, the XPO1 inhibitor, Selinexor, also showed anti-tumor activities in other human malignancies that are not regulated by AR signaling pathways. These lines of experimental and clinical evidence suggest that XPO1 inhibition is also targeting other oncogenic pathways, rather than only AR action in DNPC growth and progression. We also agree with the Reviewer that the roles of XPO1 in prostate tumor progression and DNPC development should be further investigated using the newly developed *in vivo* models. As presented in this revision, we performed a series of experiments to test pharmacological inhibition of XPO1 in PCa cells using both *ex vivo* and *in vivo* approaches, which have been included in Figs 6, 7, and Supplementary Figs. 6, 7 in the current revision. Finally, in this revision, we also modified related text (see Pages 22-23, lines 492-502) and re-organized figures to make our data clearer to the Reviewer and readers.

7. This study is mainly limited by the lack of experimental validation using human prostate cancer cell lines. I understand the author established important mouse models. At least, the author should (1) Validate the role of XPO1 in DNPC cell lines, such as PC3 and DU145, both in vitro and in vivo; (2) Use human PCa datasets to explore whether human DNPCs also highly expressed XPO1.

We appreciate the Reviewer’s point. Actually, a previous study has shown increased XPO1 expression in PC3 and DU145, AR-negative cell lines, in comparison to LNCaP, an AR positive cell line. Selective inhibition of nuclear export activity in those AR-negative prostate cell lines significantly reduced tumor cell proliferation and induced apoptosis (PMID: 26620414). Additionally, a separate study reported a decrease in tumor growth using PC3- and DU145-derived xenografts upon treatment with Selinexor alone and in combination with docetaxel (PMID: 29340049). These data demonstrated the promotional role of XPO1 on PCa growth, and also implicated that XPO1-mediated tumor growth may not be only regulated through the AR oncogenic pathways in PCa cells. In this study, we also tested XPO expression and related activation using the above human PCa cell lines as

well as DNPC-like tumor cells derived from TripleTg mice in both organoid cultures and KCT approaches. Our data from human PCa cell lines are very similar to the previous report so we did not include them here. However, in this revision, we did present our data from both organoid cultures and KCT experiments (see Figs 6-7). Moreover, we also show increased expression of XPO1 transcripts and proteins in human samples and datasets (see Supplementary Fig. 5a, b, and d).

Reviewer #4 (Remarks to the Author):

In this manuscript, the authors have addressed an important question that has been inadequately represented in the field during the last couple of years. Although almost all primary prostate cancer cells express androgen receptor and are dependent on androgens, androgen deprivation therapy fails in most patients who develop castration-resistant prostate cancer. To address this question, they carried out a series of careful and well thought out experiments involving a crop of mouse models.

The current study indicates that XPO1 inhibition, by selinexor, might be a therapeutic option for treating Double-Null Prostate Cancer. This is an important result for clinicians and researchers in the field and those working outside the field.

We greatly appreciate the Reviewer's insightful comments on the significance and clinical relevance of this study. As mentioned by the Reviewer, despite the use of more potent AR antagonists and blockers for androgen synthesis were used recently to improve clinical outcomes, more diverse CRPC phenotypes have been developed in patients treated with current ADT. In this study, we directly addressed this extremely significant challenge in the field. Using patient datasets/samples, newly developed GEMMs, and other advanced experimental approaches and analyses, we specifically investigated the pathogenesis of DNPC, which frequently occurred in CRPC patients treated with ABI and ENZ. Our data implicate a new molecular mechanism underlying current ADT induced-tumor progression and DNPC development, and provides the mechanistic insights into the development of new therapeutic strategies for treating advanced prostate cancer to delay or prevent DNPC development.

Below are some specific minor suggestions that could improve the manuscript.

We have carefully addressed each of the Reviewer's points below.

1. Page 3, line 50-52. Although the authors claim that Selinexor is FDA-approved they should cite the relevant literature.

In response to the Reviewer's point, we provided the relevant references for Selinexor in this revision (see Page 3, lines 48-50).

2. The authors demonstrate differences between Adeno-PCa and Solid-PCa cells. The Adeno-PCa cells as depicted are morphologically smaller. This fact should be addressed.

As shown in this study, "Adeno-PCa" cells maintain prostatic glandular characteristics and appear morphologically smaller than "Solid-PCa" cells. In contrast, "Solid-PCa" cells contain poorly differentiated elements with abundant and large lightly eosinophilic cytoplasm and pleomorphic nuclei without distinct gland formation. Additionally, "Solid-PCa" cells also display large and irregular nuclei with prominent and often multiple nucleoli. In response to the Reviewer, we provided more description regarding the cellular properties of both Adeno- and Solid-PCa into the "Results" section of this revision (Page 7, lines 152-54; Pages 7-8, lines 155-61).

3. *Page 6, paragraph 2. The authors elegantly demonstrate that the numbers of nuclear pore complexes differ. However no thoughts were give about the reasons and if possibly the composition of the nuclear pore complexes might be affected and potentially differ in the cell types.*

We greatly appreciate the Reviewer's point and apologize for the omission. Emerging evidence has shown the critical role of the nuclear pore complexes (NPC) functioning as the central mediators of nucleo-cytoplasmic transport. Increased numbers of NPC can amplify the nuclear transport machinery, which has been frequently observed in tumor cells. Accordingly, reduction of NPC numbers can decrease nuclear transport to induce cancer cell death. Therefore, identifying increased NPC in "Solid-PCa" cells is consistent with our data showing aberrant activation of XPO1 and ribosomal biogenesis pathways, demonstrating the link of increased nuclear transport machinery in tumor progression and DNPC development. In the current revision, we further emphasized the above points in response to the Reviewer's comment (see Page 9, lines 199-201).

4. *Figure 6. The annotation in the figure is erroneous*

Sorry for the error, and we have corrected it in the current revision.

5. *The authors should comment on the nuclear import of AR and the potential role of nuclear import in their model.*

Again, we greatly appreciate this very insightful and important point from the Reviewer. The AR is a nuclear hormone receptor, and forms a complex with heat-shock proteins (HSPs) in the cytoplasm when unbound with androgens. Upon binding to androgens, the AR dissociates from the HSPs and translocates into the nucleus. The activation of the AR through the ligand-receptor interaction is a central axis in prostate tumorigenesis. Primary prostate cancer cells express the AR and require androgens for their oncogenic growth and survival. Therefore, current ADT eliminates AR nuclear localization, blocks the ligand-receptor interaction, and inhibits AR-mediated oncogenic growth. The expression of nuclear AR appeared in Adeno-PCa cells of the TripleTg mice. However, AR nuclear localization was gradually reduced and limited, as observed in "Solid-PCa" cells during the course of tumor progression in TripleTg mice. Castration of TripleTg mice showed androgen-independent tumor growth. The previous study has shown a ligand-independent nuclear import mechanism in prostate cancer cells (PMID: 33332287). The lack of AR nuclear expression observed in "Solid-PCa" cells may be regulated through aberrant nuclear import and export pathways. Therefore, the current mouse models provide a new and relevant system for further investigating these important but unresolved questions in the field. In response to the Reviewer's comment, we provided more discussion related information for AR nuclear import and related dysregulation in the current revision (Page 22, lines 488-491, lines 491-499).

6. *The authors should discuss what is known about XPO AR interaction.*

We appreciate the Reviewer's point. Earlier studies have suggested several different mechanisms for AR nuclear import and export in prostate cancer cells. Dysregulation of XPO1 has been shown to directly and/or indirectly affect the status of nuclear AR. In this revision, we have expanded our discussion with related literature to address the reviewer's question and provided related references (Page 22, lines 491-93).

The methodology is sound and meets the recent standards. In summary, I think the paper is acceptable after carefully addressing the minor issues given above.

We thank the Reviewer again for his/her constructive comments to improve our manuscript. We sincerely believe that the current revised manuscript has precisely and adequately addressed the

Reviewers' points and emerged considerably stronger. This study contains novel, scientifically significant and rigorous information, and will greatly impact the field.

REVIEWER COMMENTS

Reviewer #1 (Remarks to the Author):

Title Androgen Deprivation Induces Double-Null Prostate Cancer via Aberrant Activation of Nuclear Export and Ribosomal Biogenesis through HGF and WNT Axes

Authors: Anonymous

Summary: This is a revised manuscript reporting a series of studies evaluating the role of HGF/MET and WNT/B-catenin signaling in the double null phenotype of prostate cancer. The work involves mining human metastatic prostate cancer datasets to identify candidate drivers of DNPC, the generation of several GEM models, the evaluation of subsequent phenotypes that included alterations in nuclear export, and the evaluation of pharmacological inhibition of nuclear export and ribosome biogenesis.

Comments:

1. The authors have been generally responsive to the critiques/comments in the initial review.
2. Overall, the manuscript is very well written and clear. The mouse model studies are solid and the data are compelling and conclusive.
3. The problems with the study continue to center on the clinical relevance and the data analyses from human databases and the biospecimens that comprise this study.
4. There is a major issue with the results shown in Figure 1a – the expression data from the SU2C/PCF RNAseq datasets. There are not 119 tumors in the cohort that are classified as DNPC. There are far less. This analysis should be carefully reanalyzed for accuracy as the original study, nor many subsequent studies using these data have classified more than 10% of the cases as DNPC. There is a problem.
5. Further, in this SU2C/PCF dataset – it does not appear that the HGF/MET and B-Catenin pathway genes are substantially different comparing NEPC and DNPC.
6. The authors should acknowledge in the results, that the number of mCRPC cases post-ABI and ENZ is only 6 cases. This is now described in the methods but needs to be clearly stated

in the results section for context. Further, the authors should note that 3 samples are from brain metastasis – a very very rare site for human prostate cancer metastasis. Further, it is not clear if these 3 brain metastases were from the same patient, or were from 3 different patients. This should be stated.

7. Unfortunately, a key requested set of experiments were not performed in the revision – these involve cause-effect experiments demonstrating that HGF/met and WNT/B-catenin signaling, when activated in AR-active human prostate cancer models, results in a DNPC phenotype with concomitant upregulation of XPO1 and differential sensitivity to XPO1 inhibition. These studies should be performed in human prostate cancer models – cell lines as the starting point, with more than one ARPC model used. These experiments are of critical importance in view of the authors contention and strong suggestion that this pathway/mechanism represents a new therapeutic target – thus promoting the design and conduct of a human clinical trial. The authors recognize that mouse models may have distinct differences from human prostate cancer biology – hence the importance to ensure rigor in the translational conclusions.

Minor Points:

1. The abstract states that “Specifically, a subpopulation of AR- and neuroendocrine (NE)-null PCa, double-null PCa (DNPC), occurs frequently in abiraterone (ABI) and enzalutamide (ENZ)-treated patients, significantly increasing PCa mortality” – the authors should be cautious with these statements. DNPC does not occur frequently in ABI or ENZ treated patients. It remains a rare entity. It is also not clear (yet) that this entity increases PCa mortality.

2. It remains unclear why, if HGF/cMET activation in the mouse prostate is inducing Wnt/b-catenin signaling, that further b-catenin manipulation is necessary to drive a DNPC phenotype?

3. It should be clarified that the mechanisms leading to DNPC in the murine vs human situation are completely different. The murine studies drive DNPC via genetic manipulation in the setting of normal androgen levels and without Ar targeting – whereas de novo DNPC is even more rare than what occurs in mCRPC – which occurs via AR pathway suppression – that latter of which is not evaluated in the mouse models.

4. It is not clear if the two DNPC human models, PC3 and DU145 cells, used for the XPO studies, exhibit high HGF/cMET and B-catenin signaling relative to ARPC and NEPC lines? Are these lines consistent with the ov

Reviewer #2 (Remarks to the Author):

The manuscript is much improved. The authors have addressed my concerns.

Reviewer #3 (Remarks to the Author):

The authors have addressed most of my concerns.

Reviewer #4 (Remarks to the Author):

The authors addressed my concerns and deeply improved the manuscript which is acceptable for publication now.

Reviewer's Comments:

Reviewer #1 (Remarks to the Author)

Title Androgen Deprivation Induces Double-Null Prostate Cancer via Aberrant Activation of Nuclear Export and Ribosomal Biogenesis through HGF and WNT Axes

Authors: Anonymous

Summary: This is a revised manuscript reporting a series of studies evaluating the role of HGF/MET and WNT/B-catenin signaling in the double null phenotype of prostate cancer. The work involves mining human metastatic prostate cancer datasets to identify candidate drivers of DNPC, the generation of several GEM models, the evaluation of subsequent phenotypes that included alterations in nuclear export, and the evaluation of pharmacological inhibition of nuclear export and ribosome biogenesis.

Comments:

1. The authors have been generally responsive to the critiques/comments in the initial review.

We appreciate the Reviewer's comment for our effort on this revision.

2. Overall, the manuscript is very well written and clear. The mouse model studies are solid and the data are compelling and conclusive.

Again, we greatly appreciate the Reviewer's positive comments on this study.

3. The problems with the study continue to center on the clinical relevance and the data analyses from human databases and the biospecimens that comprise this study.

In this revision, we have again taken the Reviewer's comments to heart and given our best effort to adequately address each of the additional questions raised by this Reviewer below. Specifically, we revised the related sentences to "tone down" the translational significance on this study (see page 2 line 41, Page 3 line 51, page 20 line 452-3, page 21, line 457, line 461; page 22 line 501).

4. There is a major issue with the results shown in Figure 1a – the expression data from the SU2C/PCF RNAseq datasets. There are not 119 tumors in the cohort that are classified as DNPC. There are far less. This analysis should be carefully reanalyzed for accuracy as the original study, nor many subsequent studies using these data have classified more than 10% of the cases as DNPC. There is a problem.

As indicated in this revised manuscript (see page 3 lines 54-56 and page 44 lines 929-30), we used the SU2C/PCF RNAseq dataset (n=266) at cBioPortal reported by Abida *et al* at 2019 (PNAS, **116**, 11428-11436, cited as Reference 11 in this revision). We defined the ARPC/NEPC/DNPC subtypes following the original report by Peter Nelson's group (see PMID: 29017058), listed as Reference 9 in this revised manuscript. The recent study by Dr. Sarki Abdulkadir's group also reported ARPC (40.1%), NEPC (10.4%), and DNPC (47.2%) subtypes using the above RNAseq databases (PMID: 35405009), which is consistent with our results reported in this study, and also directly addressed the Reviewer's question.

5. Further, in this SU2C/PCF dataset – it does not appear that the HGF/MET and B-Catenin pathway genes are substantially different comparing NEPC and DNPC.

In the same vein, we used the above datasets to further analyze the differences between the ARPC/NEPC/DNPC subtypes. As indicated in the current revision (see the modified Fig. 1b and Supplementary Fig. 1b, and page 3, lines 61-62 in the current revision), whereas a similar enrichment in

the down-regulation of androgen-response pathways was observed in DNPC and NEPC in comparison to ARPC samples, significant up-regulation of HGF/MET and canonical Wnt signaling downstream target genes was identified only between DNPC and ARPC by comparing transcriptomic changes with ARPC, but not between NEPC and ARPC samples (see Fig. 1b, Supplementary Fig. 1b, and pages 3-4, lines 63-68 in the current revision). Additionally, using GSEA, we also identified a significant enrichment in the up-regulation of HGF/MET and Wnt/ β -catenin signaling pathways using the DEGs from DNPC versus ARPC (Fig. 1b and Supplementary Fig. 1c), but not NEPC versus ARPC (Supplementary Fig. 1b). Taken together, these results directly address the Reviewer's question.

6. The authors should acknowledge in the results, that the number of mCRPC cases post-ABI and ENZ is only 6 cases. This is now described in the methods but needs to be clearly stated in the results section for context. Further, the authors should note that 3 samples are from brain metastasis – a very very rare site for human prostate cancer metastasis. Further, it is not clear if these 3 brain metastases were from the same patient, or were from 3 different patients. This should be stated.

As indicated in the current revision (Page 23, line 511-4), a total of six samples, including three metastatic brain samples that were from three individual patients, were analyzed in this study. In response to the Reviewer's comment, we also stated the information in the Results section and Figure legends (see page 4 lines 73-76, and page 43 lines 943-44).

7. Unfortunately, a key requested set of experiments were not performed in the revision – these involve cause-effect experiments demonstrating that HGF/met and WNT/ β -catenin signaling, when activated in AR-active human prostate cancer models, results in a DNPC phenotype with concomitant upregulation of XPO1 and differential sensitivity to XPO1 inhibition. These studies should be performed in human prostate cancer models – cell lines as the starting point, with more than one ARPC model used. These experiments are of critical importance in view of the authors contention and strong suggestion that this pathway/mechanism represents a new therapeutic target – thus promoting the design and conduct of a human clinical trial. The authors recognize that mouse models may have distinct differences from human prostate cancer biology – hence the importance to ensure rigor in the translational conclusions.

As described in this manuscript, based on the analyses from relevant human prostate cancer datasets, we developed a series of new GEMMs to directly assess the role of aberrant co-activation of HGF/MET and Wnt/ β -catenin in prostate cancer progression, metastasis, and DNPC development, which has been well appreciated and recognized by this and other reviewers per their previous review critiques. These new and biologically relevant models allowed us to investigate and identify the collaborative, reconcilable, and consequential effects by these different oncogenic pathways on promoting and advancing DNPC development during the course of tumor progression. Additionally, we also provided multiple lines of new and relevant scientific evidence from patient datasets and human cell lines to support our findings from our GEMMs in this revision. Moreover, we well recognized the previous studies by Dr. Peter Nelson and others using human cell lines to investigate DNPC pathogenesis, and thus have taken different approaches by devoting significant resources and spending more than three years to generate and analyze the current *in vivo* models to address this important but unclear question in the field. We thus respectfully feel that it is unnecessary for all of studies to have to “*be performed in human prostate cancer models – cell lines as the starting point.....*”. Specifically, we believe that this should at least not be used as a criterion to evaluate a scientific study. In contrast, emerging evidence has shown the significant role of tumor microenvironment in prostate cancer initiation, progression, metastasis, and CRPC development. Therefore, based on the data presented in this study, we believe that future efforts should focus on the collaborative role of those oncogenic pathways in DNPC pathogenesis using biologically relevant and *in vivo* model systems. Therefore, in this revision, we have specifically emphasized the collaborative roles and interactions between these

oncogenic pathways induced by current ADT to promote tumor progression and metastasis, and DNPC development.

Minor Points:

1. The abstract states that “Specifically, a subpopulation of AR- and neuroendocrine (NE)-null PCa, double-null PCa (DNPC), occurs frequently in abiraterone (ABI) and enzalutamide (ENZ)-treated patients, significantly increasing PCa mortality” – the authors should be cautious with these statements. DNPC does not occur frequently in ABI or ENZ treated patients. It remains a rare entity. It is also not clear (yet) that this entity increases PCa mortality.

We appreciate the Reviewer’s comment. The earlier study by Peter Nelson’s group (PMID: 29017058) reported that DNPC incidence has increased from 5.4% before FDA-approval of enzalutamide and abiraterone (years 1998–2011) to 23.3% since the approval of ENZ/abiraterone (years 2012–2016). This line of clinical evidence suggests that loss of AR expression could be due to the increased efficacy of AR antagonists. In response to the Reviewer’s comment, we modified the above sentence (see page 1, line 10, and page 2, lines 33-34).

2. It remains unclear why, if HGF/cMET activation in the mouse prostate is inducing Wnt/b-catenin signaling, that further b-catenin manipulation is necessary to drive a DNPC phenotype?

As described in this revision, we provided multiple lines of experimental evidence to demonstrate the aberrant activation of HGF/Met signaling pathways to induce Wnt/β-catenin activation (see Fig. 2 and Supplementary Fig. 2). These data provide the biological relevance, rationale, and experimental evidence for investigating the co-activation of HGF/MET and Wnt/β-catenin in prostate cancer progression and DNPC development. Moreover, one of the most important findings in this study is to identify DNPC development only in the TripleTg mice, $hHGFtg:H11^{hMET/+}:Ctnnb1^{L(Ex3)/+}:PB^{Cre4}$, and not in either the DoubleTg, $hHGFtg:H11^{hMET/+}:PB^{Cre4}$, or stabilized β-catenin only mice, $Ctnnb1^{L(Ex3)}:PB^{Cre4}$. These data directly demonstrated the critical role of co-activating HGF/MET and Wnt/β-catenin signaling pathways to promote DNPC tumorigenesis. They also directly address this Reviewer’s question.

3. It should be clarified that the mechanisms leading to DNPC in the murine vs human situation are completely different. The murine studies drive DNPC via genetic manipulation in the setting of normal androgen levels and without Ar targeting – whereas de novo DNPC is even more rare than what occurs in mCRPC – which occurs via AR pathway suppression – that latter of which is not evaluated in the mouse models.

Actually, in this study, we indeed examined the effect of castration on TripleTg mice, $hHGFtg:H11^{hMET/+}:Ctnnb1^{L(Ex3)/+}:PB^{Cre4}$. More invasive tumor lesions occurred both locally and at distant sites in castrated mice compared to age- and genotype-matched intact counterparts (see the current Fig. 7a, b). IHC further showed specific expression of pMET and nuclear β-catenin with a lack of nuclear AR and SYN expression in both prostate and lung metastatic tumor cells (Fig. 7c), confirming the double-null cell properties of mCRPC in castrated TripleTg mice. GSEA using pre-ranked gene lists from the DEGs between castrated and intact samples revealed significant enrichment in HGF and β-catenin signaling pathway activation (Fig. 7e). A significant increase in *Xpo1* expression was identified in castrated versus intact samples from TripleTg mice (Fig. 7f and Supplementary Fig. 6a). All of these data demonstrate the biological relevance of the TripleTg mice and also directly address the Reviewer’s question.

4. *It is not clear if the two DNPC human models, PC3 and DU145 cells, used for the XPO studies, exhibit high HGF/cMET and B-catenin signaling relative to ARPC and NEPC lines? Are these lines consistent with the ov*

In the literature, both PC3 and DU145 cells are AR negative prostate cancer cells lines. They have been shown to possess high HGF/cMET and Wnt/ β -catenin activity in comparison to LNCaP cells, an AR positive line, and some other AR positive prostate cancer cell lines (PMID: 12475693; PMID: 26806347; PMID: 19026633, PMID: 22820499, and PMID: 24399733). Unfortunately, the last sentence of the Reviewer appeared incomplete and unclear to us.

Reviewer #2 (Remarks to the Author)

The manuscript is much improved. The authors have addressed my concerns.

Reviewer #3 (Remarks to the Author)

The authors have addressed most of my concerns.

Reviewer #4 (Remarks to the Author)

The authors addressed my concerns and deeply improved the manuscript which is acceptable for publication now.

We truly appreciate Reviewers 2-4 constructive comments, and their time and effort to review this manuscript.

REVIEWER COMMENTS

Reviewer #1 (Remarks to the Author):

Title: Androgen Deprivation Induces Double-Null Prostate Cancer via Aberrant Activation of Nuclear Export and Ribosomal Biogenesis through HGF and WNT Axes

Authors: Anonymous

Summary: This is the second revision of a manuscript reporting a series of studies evaluating the role of HGF/MET and WNT/B-catenin signaling in the double null phenotype of prostate cancer.

Comments:

1. The authors provided responses to queries raised previously. However, they have not responded adequately and there remain: a) substantial errors in allocating 'double negative prostate cancer' (DNPC) phenotypes in human datasets and b) a lack of robust translation of the GEM model findings into human disease.
2. As noted in the prior review: "There is a major issue with the results shown in Figure 1a – the expression data from the SU2C/PCF RNAseq datasets. There are not 119 tumors in the cohort that are classified as DNPC. There are far less. This analysis should be carefully reanalyzed for accuracy as the original study, nor many subsequent studies using these data have classified more than 10% of the cases as DNPC."

The authors state that they have defined ARPC/NEPC/DNPC subtypes following a previous report (PMID: 29017058). However, a review of this paper does not identify a method whereby RNAseq and a AR/NE score or GSVA was used to classify tumor phenotypes. The initial DNPC phenotype was classified based on immunohistochemistry. Transcript-based classification in this study used MDS (multidimensional scaling) plots incorporating a panel of 10 genes regulated by the AR in prostate cancer and a panel of 10 genes associated with neuroendocrine prostate cancer. There was no explicit cut-off/cut-points for DNPC classification based on transcript levels. Overall <10% of tumors classified as DNPC (see Figure S1 in this study).

A cited study (PMID: 35405009) evaluated the SU2C RNAseq and used a classification system as follows: “To classify androgen receptor–dependent prostate cancer (ARPC, NEPC, DNPC), and ARPC/NEPC populations from the Stand Up To Cancer (SU2C) 2019 in dataset from Fig. 5A, we normalized the scores from 0 to 1 and arbitrarily set up a threshold of 0.5. Patients with AR score and NEPC scores <0.5 were classified as ARPC (n = 85). Patients with NEPC score >0.5 and AR score <0.5 were classified within the NEPC group (n = 22). Patients with DNPC had both AR score and NEPC scores <0.5 (n = 100). Finally, patients with NEPC/AR had AR and NEPC scores >0.5 (n = 5).”

Unfortunately, this is a major error in this publication and the approach incorrectly classifies DNPC tumors. Classifying an AR score <0.5 as DNPC is not accurate. Evaluating the gene expression from the majority of these ‘DNPC’ tumors shows AR expression and AR target expression. They are not DNPC. It appears that the present study used a similar approach for PC classification. While certainly this was published previously, the error should not be perpetuated and is very misleading to the scientific community.

An analysis of the same SU2C data in the cBIO portal to identify phenotypes was reported in Nyquist et al (PMID: 32460015). In this analysis, only ~10% tumors classify as DNPC based on RNAseq data.

These results are in alignment with a study by Lundberg et al (PMID: 37289025) evaluating the West Coast SU2C data (also cited in the present study – Reference 28). In this study, of 210 tumors, 13 were classified as DNPC representing only ~6 % of the cohort.

In sum, the present study misclassifies a substantial majority of the tumors called DNPC, which calls into question both the premise of developing these specific GEM models, and the applicability of the murine findings to human PC without appropriate and accurate comparisons to clinical data. Note – the models may very well be relevant for true human DNPC – but the analyses performed do not provide the adequate specificity.

3. As per the prior review: “Unfortunately, a key requested set of experiments were not performed in the revision – these involve cause-effect experiments demonstrating that

HGF/met and WNT/B-catenin signaling, when activated in AR-active human prostate cancer models, results in a DNPC phenotype with concomitant upregulation of XPO1 and differential sensitivity to XPO1 inhibition. These studies should be performed in human prostate cancer models – cell lines as the starting point, with more than one ARPC model used. These experiments are of critical importance in view of the authors contention and strong suggestion that this pathway/mechanism represents a new therapeutic target – thus promoting the design and conduct of a human clinical trial. The authors recognize that mouse models may have distinct differences from human prostate cancer biology – hence the importance to ensure rigor in the translational conclusions.”

Unfortunately, no relevant human AR-positive models were evaluated to confirm the activated pathways do indeed drive DNPC. Compared to the murine studies, these are not challenging experiments to perform. This remains a critical deficiency without which, and coupled with the errors in classifying human DNPC, the human relevance of the findings is questionable.

4. In a minor point raised regarding the development of DNPC in the absence of ADT/castration in the GEM models– it is clearly stated in the manuscript results that the TripleTg model developed a ‘solid’ phenotype lacking AR or neuroendocrine features and these features were also observed in metastasis (non-castrate). While apparently accelerated with castration - both occurred in androgen-intact mice. It should be acknowledged that while this may occur, this is an extremely rare event in human PC (the development of AR-null NE-null prostate cancer in a non-castrate human).

Reviewer #3 (Remarks to the Author):

I agree with Reviewer #1 who remind me to think about the foundational dataset used in this study. The percentage of DNPC is far less than the authors claimed here. Reviewer #1 raised very careful and important concerns. The authors must address these comments. The authors can re-analysis the dataset in a standard way or using the published dataset (<https://doi.org/10.1172/JCI128212>.) to address Reviewer #1's comments.

REVIEWER COMMENTS

Reviewer #1 (Remarks to the Author):

Title: Androgen Deprivation Induces Double-Null Prostate Cancer via Aberrant Activation of Nuclear Export and Ribosomal Biogenesis through HGF and WNT Axes

Authors: Anonymous

Summary: This is the second revision of a manuscript reporting a series of studies evaluating the role of HGF/MET and WNT/B-catenin signaling in the double null phenotype of prostate cancer.

Comments:

1. The authors provided responses to queries raised previously. However, they have not responded adequately and there remain: a) substantial errors in allocating 'double negative prostate cancer' (DNPC) phenotypes in human datasets and b) a lack of robust translation of the GEM model findings into human disease.

We appreciate the Reviewer's comments. Again, we have taken them into our heart and fully address them below.

2. As noted in the prior review: "There is a major issue with the results shown in Figure 1a – the expression data from the SU2C/PCF RNAseq datasets. There are not 119 tumors in the cohort that are classified as DNPC. There are far less. This analysis should be carefully reanalyzed for accuracy as the original study, nor many subsequent studies using these data have classified more than 10% of the cases as DNPC."

The authors state that they have defined ARPC/NEPC/DNPC subtypes following a previous report (PMID: 29017058). However, a review of this paper does not identify a method whereby RNAseq and a AR/NE score or GSVA was used to classify tumor phenotypes. The initial DNPC phenotype was classified based on immunohistochemistry. Transcript-based classification in this study used MDS (multidimensional scaling) plots incorporating a panel of 10 genes regulated by the AR in prostate cancer and a panel of 10 genes associated with neuroendocrine prostate cancer. There was no explicit cut-off/cut-points for DNPC classification based on transcript levels. Overall <10% of tumors classified as DNPC (see Figure S1 in this study).

A cited study (PMID: 35405009) evaluated the SU2C RNAseq and used a classification system as follows: "To classify androgen receptor-dependent prostate cancer (ARPC, NEPC, DNPC), and ARPC/NEPC populations from the Stand Up To Cancer (SU2C) 2019 in dataset from Fig. 5A, we normalized the scores from 0 to 1 and arbitrarily set up a threshold of 0.5. Patients with AR score and NEPC scores <0.5 were classified as ARPC (n = 85). Patients with NEPC score >0.5 and AR score <0.5 were classified within the NEPC group (n = 22). Patients with DNPC had both AR score and NEPC scores <0.5 (n = 100). Finally, patients with NEPC/AR had AR and NEPC scores >0.5 (n = 5)." Unfortunately, this is a major error in this publication and the approach incorrectly classifies DNPC tumors. Classifying an AR score <0.5 as DNPC is not accurate. Evaluating the gene expression from the majority of these 'DNPC' tumors shows AR expression and AR target expression. They are not DNPC. It appears that the present study used a similar approach for PC classification. While certainly this was published previously, the error should not be perpetuated and is very misleading to the scientific community.

An analysis of the same SU2C data in the cBIO portal to identify phenotypes was reported in Nyquist et al (PMID: 32460015). In this analysis, only ~10% tumors classify as DNPC based on

RNAseq data. These results are in alignment with a study by Lundberg et al (PMID: 37289025) evaluating the West Coast SU2C data (also cited in the present study – Reference 28). In this study, of 210 tumors, 13 were classified as DNPC representing only ~6 % of the cohort.

In sum, the present study misclassifies a substantial majority of the tumors called DNPC, which calls into question both the premise of developing these specific GEM models, and the applicability of the murine findings to human PC without appropriate and accurate comparisons to clinical data. Note – the models may very well be relevant for true human DNPC – but the analyses performed do not provide the adequate specificity.

We appreciate the Reviewer’s additional points regarding the data in Fig. 1a related to the classification of DNPC tumors. We also appreciate the Reviewer’s comments on the previous study by Rodriguez *et al.* (PMID: 35405009). To directly and specifically address the Reviewer’s comments in the revision, we re-analyzed the same SU2C data in the cBIO portal following the report by Nyquist *et al.* (PMID: 32460015) as the Reviewer suggested. We obtained the same number (14 samples; 4%) of DNPC as reported previously (PMID: 32460015). Importantly, analysis of these DNPC samples also showed significant activation of HGF/Met and Wnt/ β -catenin pathways in comparison to ARPC samples. These data directly address the Reviewer’s concern and provide the scientific evidence demonstrating aberrant activation of HGF/Met and Wnt/ β -catenin pathways in DNPC cells. They are also consistent with the data from our IHC and scRNA-seq analyses with human samples and datasets and directly support the scientific promise of this study. We revised Fig. 1a and other related figures with these new lines of results and also modified related text in this revision (please see Page 3, lines 54-56; lines 60-62; lines 63-65; Page 4, line 67; lines 68-72; Page 18, lines 396-97; Pages 23-24, lines 515-22; Page 44, lines 934-40).

3. As per the prior review: “Unfortunately, a key requested set of experiments were not performed in the revision – these involve cause-effect experiments demonstrating that HGF/met and WNT/B-catenin signaling, when activated in AR-active human prostate cancer models, results in a DNPC phenotype with concomitant upregulation of XPO1 and differential sensitivity to XPO1 inhibition. These studies should be performed in human prostate cancer models – cell lines as the starting point, with more than one ARPC model used. These experiments are of critical importance in view of the authors contention and strong suggestion that this pathway/mechanism represents a new therapeutic target – thus promoting the design and conduct of a human clinical trial. The authors recognize that mouse models may have distinct differences from human prostate cancer biology – hence the importance to ensure rigor in the translational conclusions.”

Unfortunately, no relevant human AR-positive models were evaluated to confirm the activated pathways do indeed drive DNPC. Compared to the murine studies, these are not challenging experiments to perform. This remains a critical deficiency without which, and coupled with the errors in classifying human DNPC, the human relevance of the findings is questionable.

As explained in the revised manuscript and our early responses to the Reviewer, while we appreciate the Reviewer’s point, we also recognize the limitations of the current available human ARPC cell lines and related *in vitro* experimental approaches. Additionally, in response to the Reviewer, we have modified the manuscript to tone down our conclusions in order to retain rigor in the translational studies in the previous revision.

4. In a minor point raised regarded the development of DNPC in the absence of ADT/castration in the GEM models– it is clearly stated in the manuscript results that the TripleTg model developed a ‘solid’ phenotype lacking AR or neuroendocrine features and these features were also observed in metastasis (non-castrate). While apparently accelerated with castration - both occurred in androgen-intact mice.

It should be acknowledged that while this may occur, this is an extremely rare event in human PC (the development of AR-null NE-null prostate cancer in a non-castrate human).

We appreciate the Reviewer's comment. As indicated in our revised manuscript, to mimic an increase in HGF and Wnt signaling in DNPC, we developed TripleTg mice to directly examine their role in prostate tumorigenesis. These mice showed the aggressive local and metastatic tumor phenotypes, pathology very similar to DNPC, and resistance to androgen withdrawal. We agree that it may be very rare for prostate cancer patients to develop DNPC without ADT. However, given the current data showing aberrant activation of HGF/Met and Wnt/ β -catenin pathways in DNPC patients, the current mouse models can be used to investigate the role of HGF/Met and Wnt/ β -catenin pathways in DNPC or DNPC-like tumor development, which is the true purpose for the development of these *in vivo* models.

Reviewer #3 (Remarks to the Author):

I agree with Reviewer #1 who remind me to think about the foundational dataset used in this study. The percentage of DNPC is far less than the authors claimed here. Reviewer #1 raised very careful and important concerns. The authors must address these comments. The authors can re-analyze the dataset in a standard way or using the published dataset (<https://doi.org/10.1172/JCI128212>.) to address Reviewer #1's comments.

As we responded to Reviewer 1 above, we re-analyzed the same SU2C dataset in the cBIO portal following the report by Nyquist *et al.* (PMID: 32460015). We observed aberrant activation of HGF/Met and Wnt/ β -catenin pathways in newly classified DNPC samples in comparison to ARPC samples, which directly address Reviewer 1's concerns. We provided these new data in the current revision (please see the Revised Fig. 1a-d, Supplemental Fig. 1a-c, and Supplementary_Data1), and appreciate the Reviewer's time and comment again.

REVIEWERS' COMMENTS

Reviewer #1 (Remarks to the Author):

I appreciate that the authors have revised the analysis of the prostate cancer datasets and the classification of DNPC is now more in alignment with prior results from these datasets. It should not have required a reviewer to point this out – the authors should have recognized this initially. While now more accurate, the results call into question the prior analyses which classified many ARPCs as DNPCs and consequently also had the WNT and HGF signatures enriched in these ARPC (classified as DNPC) tumors.

It is my opinion that the requested human cell experiments are important to perform and provides the needed translation of mouse models to human disease. It is the editor's decision whether the manuscript is acceptable as it is, or whether these data are sufficiently important to require inclusion. In my opinion, they are needed – but it is up to the editor and the other reviewers.

Reviewer #3 (Remarks to the Author):

The conclusion must be validated in at least one relevant human AR-positive prostate cancer model to confirm the activated pathways (HGF and Wnt) in driving the formation of DNPC.

REVIEWER COMMENTS

Reviewer #1 (Remarks to the Author):

I appreciate that the authors have revised the analysis of the prostate cancer datasets and the classification of DNPC is now more in alignment with prior results from these datasets. It should not have required a reviewer to point this out – the authors should have recognized this initially. While now more accurate, the results call into question the prior analyses which classified many ARPCs as DNPCs and consequently also had the WNT and HGF signatures enriched in these ARPC (classified as DNPC) tumors.

It is my opinion that the requested human cell experiments are important to perform and provides the needed translation of mouse models to human disease. It is the editor's decision whether the manuscript is acceptable as it is, or whether these data are sufficiently important to require inclusion. In my opinion, they are needed – but it is up to the editor and the other reviewers.

We greatly appreciate the Reviewer's effort to review this manuscript. As the Reviewer indicated, we have revised the analyses to follow his/her suggestion in this current revision. Additionally, we also provided several lines of experimental evidence demonstrating aberrant activation of HGF/Met and Wnt/ β -catenin signaling pathways and elevated expression of XPO1 and ribosomal proteins in human samples and datasets (please see Fig. 1, Fig. 5d, e, Supplementary Fig. 1, and Supplementary Fig. 5a-d). We also assessed XPO1 expression and related activation using the human PCa cell lines as we addressed in the initial review (see Supplementary Fig. 5a, b, and d). Moreover, in response to the Reviewer's point, we have modified the manuscript to tone down our conclusions in order to retain rigor in the translational studies in both this and previous revisions.

Reviewer #3 (Remarks to the Author):

The conclusion must be validated in at least one relevant human AR-positive prostate cancer model to confirm the activated pathways (HGF and Wnt) in driving the formation of DNPC.

Again, as responded above, we have provided several lines of experimental evidence demonstrating aberrant activation of HGF/Met and Wnt/ β -catenin signaling pathways in human DNPC samples and datasets. Additionally, as we indicated in the previous responses, the previous studies have shown that both PC3 and DU145 cells, AR negative prostate cancer cells lines, possess high HGF/MET and Wnt/ β -catenin activity in comparison to LNCaP cells, an AR positive line (PMID: 12475693; PMID: 26806347; PMID: 19026633, PMID: 22820499, and PMID: 24399733). Our previous data were very similar to the above findings. The above points have been addressed previously in our point-by-point responses (see minor point #4 to Reviewer 1). This Reviewer also agreed with our previous responses to the above questions in the initial revision.